# CAUTIOUS OPTIMIZERS: IMPROVING TRAINING WITH ONE LINE OF CODE

**Kaizhao Liang**[*,1,†], **Lizhang Chen**[*,1,†], **Bo Liu**[1], **Qiang Liu**[1]
[1]The University of Texas at Austin

## ABSTRACT

AdamW has been the default optimizer for transformer pretraining. For many years, our community searched for faster and more stable optimizers with only constrained positive outcomes. In this work, we propose a **one-line modification in Pytorch** to any momentum-based optimizer, which we rename cautious optimizer, e.g. C-AdamW and C-Lion. Our theoretical result shows that this modification preserves Adam's Hamiltonian function and it does not break the convergence guarantee under the Lyapunov analysis. In addition, a whole new family of optimizers is revealed by our theoretical insight. Among them, we pick the simplest one for empirical experiments, showing not only consistent speed-up on LLM pretraining, but also image classification, with minimum extra tuning on hyperparameters.

## 1 INTRODUCTION

**Algorithm 1** Caution an Optimizer (OPT) in PyTorch

```
# param p, update u from OPT, grad g
m = (u * g > 0).to(g.dtype)
p.add_(u * m/(m.mean()+eps), alpha=-lr)
```

Optimization is an important and constantly evolving field in modern machine learning. Undoubtedly, Adam (Kingma, 2014) and AdamW (Loshchilov, 2017) are the most consequential optimizers proposed almost a decade ago. Since then, many efforts (Zhang et al., 2021; Loshchilov et al., 2017) have been made to discover better and faster optimizers beyond these two. However, until now, AdamW remains the dominant workhorse for applications, from pre-training Large Language Models (LLMs) (Touvron et al., 2023) to fine-tuning text to image diffusion (Rombach et al., 2022), with no real challenges to their ruling status.

In the dawn of the era of LLMs, the arms race of model scaling intensifies (Achiam et al., 2023). A faster optimizer means more training tokens can be consumed within the same amount of time. Ultimately, this leads to more capable models (Kaplan et al., 2020). Hence, the interest in searching for an optimizer beyond AdamW is re-kindled. Recent progress in new AdamW alternatives such as Lion (Chen et al., 2024; 2023a), SHAMPOO (Gupta et al., 2018), SOAP (Vyas et al., 2024), ADOPT (Taniguchi et al., 2024), and Schedule-Free (Defazio et al., 2024), all claim substantial improvement over AdamW.

However, these methods normally require non-trivial efforts to obtain optimal results, especially hyperparameter tuning, which greatly limits their potential and wide adoption. In light of this dilemma, we propose *cautious optimizers*, an exceptionally simple performance booster of any momentum-based optimizer that only requires one line of modification (see Algorithm 1). The change is simple: *do not update unless the proposed update direction and the current gradients are aligned*. With this minor change, we obtain consistent improvement over the base optimizer without modification of the original optimal hyperparameters.

To provide an overview of the idea, let us consider a general optimizer for minimizing the loss $\mathcal{L}(\boldsymbol{w})$:

$$\boldsymbol{w}_{t+1} \leftarrow \boldsymbol{w}_t - \epsilon_t \boldsymbol{u}_t,$$

---

[*]Equal contribution by KZ and LZ. [†]Correspondence: `kaizhaol, lzchen@utexas.edu`

where $\boldsymbol{u}_t$ is the negative update direction of the parameter $\boldsymbol{w}_t$ at iteration $t$, and $\epsilon_t > 0$ is the step size. We will assume that the update represents a generic momentum-based optimizer, including, for example, Polyak and Nesterov momentum, Adam, and Lion. In all these momentum-based optimizers, $\boldsymbol{u}_t$ does not necessarily align with the gradient direction $\boldsymbol{g}_t = \nabla \mathcal{L}(\boldsymbol{w}_t)$, which can result in a temporary increase in the loss function and slow down convergence.

Cautious optimizers avoid this issue by adding a simple mask function based on the sign consistency of $\boldsymbol{u}_t$ and $\boldsymbol{g}_t$:

$$\boldsymbol{w}_{t+1} \leftarrow \boldsymbol{w}_t - \epsilon_t \boldsymbol{u}_t \circ \phi(\boldsymbol{u}_t \circ \boldsymbol{g}_t),$$

where $\circ$ denotes an element-wise product, and $\phi$ is a map that reweights the update based on the product $\boldsymbol{u}_t \circ \boldsymbol{g}_t$. We simply take it as

$$\phi(\boldsymbol{v}x) = \alpha(\boldsymbol{v}x)\mathbb{I}(\boldsymbol{v}x > 0),$$

so that the update is zeroed out for coordinates on which the sign of $\boldsymbol{u}_t$ and $\boldsymbol{g}_t$ are inconsistent. Here $\alpha(\boldsymbol{x})$ is a positive scaling factor, introduced to compensate the decrease of update magnitude due to masking. A simple choice of $\alpha$ is

$$\alpha(\boldsymbol{x}) = \frac{\texttt{dim}(\boldsymbol{v}x)}{\texttt{nnz}(\boldsymbol{v}x > 0) + \xi}, \tag{1}$$

where $\texttt{dim}(\cdot)$ and $\texttt{nnz}(\cdot)$ represent the total number of elements and the number of non-zero elements of the input vector, respectively. Here, $\xi > 0$ is a positive constant, which we set to $\xi = 1$ by default. See Algorithms 1 and 2 for more details.

This modification ensures the new negative update to have a non-negative inner product with the gradient, and hence decreases the loss monotonically when the step size is sufficiently small. Specifically, Taylor approximation shows

$$\mathcal{L}(\boldsymbol{w}_{t+1}) - \mathcal{L}(\boldsymbol{w}_t) \approx -\epsilon_t (\boldsymbol{u}_t \circ \boldsymbol{g}_t)^\top \phi(\boldsymbol{u}_t \circ \boldsymbol{g}_t) \leq 0.$$

This ensures decrease of the loss, i.e., $\mathcal{L}(\boldsymbol{w}_{t+1}) \leq \mathcal{L}(\boldsymbol{w}_t)$, when the step size is sufficiently small. In comparison, typical momentum-based optimizers do not always guarantee a monotonic decrease in loss even with infinitesimal step sizes. The nature of momentum dynamics introduces oscillations due to inertia-like effects.

Our theoretical analysis shows that the modified algorithm converges to local optima under mild conditions on the base optimizers. An interesting aspect of this algorithm is that it *does not* get stuck at non-stationary points of the loss, even if $\boldsymbol{u}_t$ can be temporarily completely conflicting with $\boldsymbol{g}_t$ and is therefore entirely masked out. This is because, in typical momentum methods, the update direction $\boldsymbol{u}_t$ continues to accumulate the gradients and will eventually be updated to have a positive inner product with $\boldsymbol{g}_t$ if it is stuck at a non-stationary point.

Our theoretical analysis encompasses both continuous-time and discrete-time cases. In the continuous-time setting, our theory shows that the modified algorithm guarantees convergence to local optima for optimizers that admit a Hamiltonian+Descent structure (Chen et al., 2023a; Liang et al., 2024), which broadly includes almost all existing popular algorithms, such as Adam, Lion, heavy ball, and Nesterov momentum. For these algorithms, we demonstrate that the cautious optimizers, with $\phi$ satisfying $x\phi(x) \geq \max(x, 0)$, retains the monotonic decreasing properties of the original Lyapunov (or Hamiltonian) functions of these algorithms while additionally minimizing the loss function. In the discrete-time setting, we analyze the behavior of various variants of mask functions and establish general conditions under which updates from cautious optimizers yield larger local descents compared to their base optimizers.

To summarize our contributions, we present the following:

- We propose Cautious Optimizers, a simple performance booster for any momentum-based optimizer, implemented with just a single line of code.
- We theoretically demonstrate that cautious optimizers preserve the convergence guarantees of the base optimizer while also accelerating the decrease of the loss function.
- We show consistent improvements across various tasks, from pretraining LLM to image classification.

---

**Algorithm 2** Cautious AdamW (C-AdamW)

---

**Require:** parameter $w$, step sizes $\{\epsilon_t\}$, dampening factors $\beta_1, \beta_2 \in [0, 1)$, $e > 0$, weight decay $\gamma \geq 0$.
1: Initialize $t = 0$, and $m_0, v_0$.
2: **while** $w_t$ not converged **do**
3: $\quad t \leftarrow t + 1$
4: $\quad g_t \leftarrow \nabla_w \mathcal{L}_t(w_{t-1})$
5: $\quad m_t \leftarrow \beta_1 m_{t-1} + (1 - \beta_1) g_t$
6: $\quad v_t \leftarrow \beta_2 v_{t-1} + (1 - \beta_2) g_t^2$
7: $\quad \hat{m}_t \leftarrow m_t / (1 - \beta_1^t)$
8: $\quad \hat{v}_t \leftarrow v_t / (1 - \beta_2^t)$
9: $\quad u_t \leftarrow \hat{m}_t / (\sqrt{\hat{v}_t} + e)$
10: $\quad \phi_t \leftarrow \mathbb{I}(u_t \circ g_t > 0)$          // Compute alignment mask
11: $\quad \bar{\epsilon}_t = \epsilon_t \frac{d}{\|\phi_t\|_0 + 1}$          // Scale lr, $d$ is dimension of $\phi_t$
12: $\quad w_t \leftarrow w_{t-1} - \bar{\epsilon}_t \phi_t \circ u_t$     // Masked update
13: $\quad w_t \leftarrow w_t - \bar{\epsilon}_t \gamma w_t$     // Add weight decay
14: **end while**

---

## 2 THEORY

We start with introducing a general Hamiltonian descent framework for the continuous-time forms of general momentum algorithms (Section 2.1). We then introduce the cautious optimizers in the continuous time form and discuss its theoretical properties (Section 2.2). Finally, we discuss in Section 2.3 theoretical properties of cautious optimizers in discrete time forms.

### 2.1 HAMILTONIAN+DESCENT

In the continuous-time form, most momentum-based algorithms can be viewed as variants of the damped Hamiltonian system, which admit a Lyapunov (or Hamiltonian) function that certifies their convergence towards the stationary points.

The Lyapunov function is typically an augmented loss function $\mathcal{H}(w, s)$ defined over both the weights $w$ and an optimization state vector $s$ (which includes the momentum). It must satisfy $\min_s \mathcal{H}(w, s) = \mathcal{L}(w)$, so that minimizing $\mathcal{L}(w)$ is equivalent to minimizing $\mathcal{H}(w, s)$. This is typically achieved using a separable Hamiltonian of the form

$$\mathcal{H}(w, s) = \mathcal{L}(w) + \mathcal{K}(s),$$

where $\mathcal{K}(\cdot)$ is any lower-bounded function. Consulting physical intuitions, we can think of $\mathcal{H}$ as the total energy of a system parameterized by $(w, s)$, with $\mathcal{L}$ and $\mathcal{K}$ representing the potential energy and kinetic energy, respectively.

The continuous-time form of common momentum-based algorithms can be unified into:

$$\begin{aligned} \frac{\mathrm{d}}{\mathrm{d}t} w_t &= -\nabla \mathcal{K}(s_t) - \Phi_t(\nabla \mathcal{L}(w_t)) \\ \frac{\mathrm{d}}{\mathrm{d}t} s_t &= \nabla \mathcal{L}(w_t) - \Psi_t(\nabla \mathcal{K}(s_t)), \end{aligned} \tag{2}$$

where $\Phi_t(\cdot)$ and $\Psi_t(\cdot)$ are *monotonic* mappings satisfying

$$\|x\|_{\Phi_t}^2 := \langle x, \Phi_t(x) \rangle \geq 0, \qquad \|x\|_{\Psi_t}^2 := \langle x, \Psi_t(x) \rangle \geq 0,$$

for any $x$. With $\Phi(x) = \Psi(x) = 0$, the system in (2) reduces to the standard Hamiltonian system, which preserves $\mathcal{H}(w_t, s_t) = \text{const}$ along the trajectory. When the descending components $\Phi_t$ and $\Psi_t$ are added, the system ensures that $\mathcal{H}(w, s)$ becomes monotonically non-decreasing:

$$\frac{\mathrm{d}}{\mathrm{d}t} \mathcal{H}(w_t, s_t) = -\Delta_{\mathcal{H}}(vw_t, vs_t) \leq 0, \tag{3}$$

$$\text{where} \quad \Delta_{\mathcal{H}}(vw_t, vs_t) := \|\nabla \mathcal{L}(w_t)\|_{\Phi_t}^2 + \|\nabla \mathcal{K}(s_t)\|_{\Psi_t}^2.$$

On the other hand, $\mathcal{L}(\boldsymbol{w})$, which is the true objective, is not necessarily decreasing monotonically. There can be cases where $\mathcal{L}(\boldsymbol{w})$ increases temporarily in exchange for a large decrease in the kinetic energy $\mathcal{K}(\boldsymbol{s})$, while still ensuring a decrease in the total energy $\mathcal{H} = \mathcal{L} + \mathcal{K}$. Specifically,

$$\frac{\mathrm{d}}{\mathrm{d}t}\mathcal{L}(\boldsymbol{w}_t) = -\Delta_{\mathcal{L}}(\boldsymbol{w}_t, \boldsymbol{s}_t),$$

$$\Delta_{\mathcal{L}}(\boldsymbol{w}_t, \boldsymbol{s}_t) \coloneqq \nabla\mathcal{L}(\boldsymbol{w}_t)^\top \nabla\mathcal{K}(\boldsymbol{s}_t) + \|\nabla\mathcal{L}(\boldsymbol{w}_t)\|_{\Phi_t}^2 . \tag{4}$$

Here, $\Delta_{\mathcal{L}}(\boldsymbol{w}_t, \boldsymbol{s}_t)$ may be negative due to the cross term.

See Appendix for the Hamiltonian of common optimizers including Adam (Kingma, 2014) and Lion-$\mathcal{K}$ (Chen et al., 2023b;a).

## 2.2 Cautious Dynamics

Our idea is to change the dynamics to make it *simultaneously* decrease both $\mathcal{H}(\boldsymbol{w}, \boldsymbol{s})$ and $\mathcal{L}(\boldsymbol{w})$. We do this with a modified system:

$$\overline{\boldsymbol{x}}_t = \nabla\mathcal{L}(\overline{\boldsymbol{w}}_t) \circ \nabla\mathcal{K}(\overline{\boldsymbol{s}}_t)$$

$$\frac{\mathrm{d}}{\mathrm{d}t}\overline{\boldsymbol{w}}_t = -\phi(\overline{\boldsymbol{x}}_t) \circ \nabla\mathcal{K}(\overline{\boldsymbol{s}}_t) - \Phi_t(\nabla\mathcal{L}(\overline{\boldsymbol{w}}_t)) \tag{5}$$

$$\frac{\mathrm{d}}{\mathrm{d}t}\overline{\boldsymbol{s}}_t = \nabla\mathcal{L}(\overline{\boldsymbol{w}}_t) - \Psi_t(\nabla\mathcal{K}(\overline{\boldsymbol{s}}_t)),$$

where $\circ$ denotes the element-wise product and $\phi$ is a vector to vector mapping. Here we weigh each element of the update direction $\nabla\mathcal{K}(\overline{\boldsymbol{s}}_t)$ based on the product of $\nabla\mathcal{K}(\boldsymbol{s})$ with the gradient $\nabla\mathcal{L}(\boldsymbol{w})$. Note that we do not need to apply a mask on the $\Phi_t(\nabla\mathcal{L}(\overline{\boldsymbol{w}}_t)$ term since it is always non-increasing by definition of $\Phi_t$.

The following conditions on the choice of function $\phi$ ensure that the system decreases *both* $\mathcal{H}$ and $\mathcal{L}$ simultaneously.

**Theorem 2.1.** *Following the dynamics in* (5) *in* $\mathbb{R}^d$*, we have*

$$\frac{\mathrm{d}}{\mathrm{d}t}\mathcal{H}(\overline{\boldsymbol{w}}_t, \overline{\boldsymbol{s}}_t) = (\overline{\boldsymbol{x}}_t^\top (\mathbf{1} - \phi(\overline{\boldsymbol{x}}_t))) - \Delta_{\mathcal{H}_t}(\overline{\boldsymbol{w}}_t, \overline{\boldsymbol{s}}_t),$$

$$\frac{\mathrm{d}}{\mathrm{d}t}\mathcal{L}(\overline{\boldsymbol{w}}_t) = -\overline{\boldsymbol{x}}_t^\top \phi(\overline{\boldsymbol{x}}_t) - \|\nabla\mathcal{L}(\overline{\boldsymbol{w}}_t)\|_{\Phi_t}^2$$

$$= (\overline{\boldsymbol{x}}_t^\top (\mathbf{1} - \phi(\overline{\boldsymbol{x}}_t))) - \Delta_{\mathcal{L}_t}(\overline{\boldsymbol{w}}_t, \overline{\boldsymbol{s}}_t),$$

*Here,* $\Delta_{\mathcal{H}_t}(\overline{\boldsymbol{w}}_t, \overline{\boldsymbol{s}}_t)$ *and* $\Delta_{\mathcal{L}_t}(\overline{\boldsymbol{w}}_t, \overline{\boldsymbol{s}}_t)$*, as defined in* (3) *and* (4)*, respectively, represent the decreasing rates of* $\mathcal{H}$ *and* $\mathcal{L}$ *in accordance with the original system* (2)*. Hence:*

• *If* $\boldsymbol{x}^\top (\mathbf{1} - \phi(\boldsymbol{x})) \leq 0$ *for any* $\boldsymbol{x} \in \mathbb{R}^d$*, then both* $\mathcal{H}$ *and* $\mathcal{L}$ *decreases faster than the original system:*

$$\frac{\mathrm{d}}{\mathrm{d}t}\mathcal{H}(\overline{\boldsymbol{w}}_t, \overline{\boldsymbol{s}}_t) \leq -\Delta_{\mathcal{H}_t}(\overline{\boldsymbol{w}}_t, \overline{\boldsymbol{s}}_t) \leq 0,$$

$$\frac{\mathrm{d}}{\mathrm{d}t}\mathcal{L}(\overline{\boldsymbol{w}}_t) \leq -\Delta_{\mathcal{L}_t}(\overline{\boldsymbol{w}}_t, \overline{\boldsymbol{s}}_t).$$

• *If* $\boldsymbol{x}^\top \phi(\boldsymbol{x}) \geq 0$ *for any* $\boldsymbol{x} \in \mathbb{R}^d$*, then* $\mathcal{L}$ *decreases monotonically,* $\frac{\mathrm{d}}{\mathrm{d}t}\mathcal{L}(\overline{\boldsymbol{w}}_t) \leq 0$.

One sufficient condition for $\phi$ to satisfy both conditions in Theorem 2.1 is to enforce the following element-wise constraint:

$$\phi(\boldsymbol{v}\boldsymbol{x})_i \geq 1 \text{ if } x_i > 0, \qquad \text{and} \qquad \phi(\boldsymbol{v}\boldsymbol{x})_i \leq 0 \text{ if } x_i < 0, \tag{6}$$

where $\phi(\boldsymbol{v}\boldsymbol{x})_i$ denotes the $i$-th element of $\phi(\boldsymbol{v}\boldsymbol{x})$. Under this condition, both $\mathcal{H}$ and $\mathcal{L}$ decrease monotonically following the cautious dynamics, at a rate faster than the original systems. In particular, the default choice $\phi(\boldsymbol{v}\boldsymbol{x}) = \alpha(\boldsymbol{v}\boldsymbol{x})\mathbb{I}(\boldsymbol{v}\boldsymbol{x} \geq 0)$ with $\alpha(\boldsymbol{v}\boldsymbol{x}) \geq 1$ satisfies the conditions above.

**Convergence to Stationary Points**  In addition to monotonically decreasing the loss function, we want to ensure that the algorithm does not get stuck unless the solution reaches a stationary point, which in practice is typically a local optimum, of the loss function. The following result demonstrates that this property holds under the same conditions as the original Hamiltonian descent system.

**Corollary 2.2.** *Assume that the norm $\| \cdot \|_\Psi^2$ is positive definite, $\Psi(0) = 0$, and that $\mathcal{H}(\boldsymbol{w}, \boldsymbol{s}) = \mathcal{L}(\boldsymbol{w}) + \mathcal{K}(\boldsymbol{s})$ is differentiable. Then, the bounded solutions of the original system (2) converge to a stationary point of $\mathcal{H}(\boldsymbol{w}, \boldsymbol{s})$. Similarly, the bounded solutions of (5) also converge to a stationary point of $\mathcal{H}(\boldsymbol{w}, \boldsymbol{s})$.*

### 2.3 DISCRETE-TIME ANALYSIS

We analyze the discrete time case, demonstrating that each step of cautious optimizers are at least as good as the step of the original optimizers under mild conditions.

We will consider a generic update of form

$$\begin{aligned}
\boldsymbol{w}_{k+1} &= \boldsymbol{w}_k - \epsilon_k \boldsymbol{u}_k(\boldsymbol{w}_k, \boldsymbol{s}_k), \\
\boldsymbol{s}_{k+1} &= \boldsymbol{s}_k + \boldsymbol{v}_k(\boldsymbol{w}_k, \boldsymbol{s}_k),
\end{aligned} \tag{7}$$

where $\boldsymbol{u}_k, \boldsymbol{v}_k$ are vector fields that define the updates, and $\epsilon_k$ is the step size. We write the cautious variants as

$$\begin{aligned}
\overline{\boldsymbol{w}}_{k+1} &= \overline{\boldsymbol{w}}_k - \epsilon_k \overline{\boldsymbol{u}}_k \circ \overline{\boldsymbol{v\phi}}_k, \quad \overline{\boldsymbol{u}}_k = \boldsymbol{u}_k(\overline{\boldsymbol{w}}_k, \overline{\boldsymbol{s}}_k) \\
\overline{\boldsymbol{s}}_{k+1} &= \overline{\boldsymbol{s}}_k + \boldsymbol{v}_k(\overline{\boldsymbol{w}}_k, \overline{\boldsymbol{s}}_k),
\end{aligned} \tag{8}$$

where $\overline{\boldsymbol{v\phi}}_k$ is a mask vector determined by the algorithm, $\overline{\boldsymbol{v\phi}}_k = \alpha(\overline{\boldsymbol{u}}_k \circ \overline{\boldsymbol{vg}}_k)\mathbb{I}(\overline{\boldsymbol{u}}_k \circ \overline{\boldsymbol{vg}}_k > 0)$, where $\overline{\boldsymbol{vg}}_k = \nabla \mathcal{L}(\overline{\boldsymbol{w}}_k)$.

The following is a comparison result showing that each step of the cautious optimizer yields larger loss decrease than the original optimizer under mild conditions.

**Theorem 2.3.** *Consider (7) and (8) with a $\mu$-smooth loss function $\mathcal{L}(\cdot)$. Assume the element-wise operator $\phi$ satisfies*

$$\Delta(\boldsymbol{vx}) := -\boldsymbol{vx}^\top (1 - \phi(\boldsymbol{vx})) \geq 0.$$

*Starting from $(\boldsymbol{w}_t, \boldsymbol{s}_t) = (\overline{\boldsymbol{w}}_t, \overline{\boldsymbol{s}}_t)$, we have*

$$\mathcal{L}(\overline{\boldsymbol{w}}_{t+1}) \leq \mathcal{L}(\boldsymbol{w}_{t+1}),$$

*which holds for step size $\epsilon_t \leq \frac{2\Delta(\overline{\boldsymbol{vu}}_t \circ \overline{\boldsymbol{vg}}_t)}{\mu \|\overline{\boldsymbol{vr}}_t\|(2 \cdot \|\overline{\boldsymbol{u}}_t\| + \|\overline{\boldsymbol{vr}}_t\|)}$, where $\overline{\boldsymbol{vr}}_t = \overline{\boldsymbol{u}}_t \circ (1 - \overline{\boldsymbol{v\phi}}_t)$ and $\overline{\boldsymbol{vg}}_t = \nabla \mathcal{L}(\overline{\boldsymbol{w}}_t)$.*

This result works only for a range of step sizes due to the need of Taylor approximation. In Appendix A.5, we show a case when the comparison holds for all step sizes when using a different mask function based on inner products and when $\mathcal{L}$ is convex. The following is another result showing that when imposing more restrictive conditions on the mask to the step size that ensure that the cautious optimizer is guaranteed to decrease loss at each step.

**Theorem 2.4.** *Consider updates (7) and (8). Assuming $\mathcal{L}(\cdot)$ is $\mu$-smooth, and consider the following mask function:*

$$\overline{\boldsymbol{v\phi}}_k = \alpha_k \mathbb{I}\left(\nabla \mathcal{L}(\overline{\boldsymbol{w}}_k) \circ \overline{\boldsymbol{vu}}_k \geq \frac{\mu\sigma}{2}\overline{\boldsymbol{vu}}_k \circ \overline{\boldsymbol{vu}}_k\right),$$

*where $\{\alpha_k\}$ is any sequence and $\sigma \geq \epsilon_k \alpha_k$. Starting from $(\boldsymbol{w}_t, \boldsymbol{s}_t) = (\overline{\boldsymbol{w}}_t, \overline{\boldsymbol{s}}_t)$, we have*

$$\mathcal{L}(\overline{\boldsymbol{w}}_{k+1}) \leq \mathcal{L}(\overline{\boldsymbol{w}}_k).$$

The results above demonstrate that cautious optimizers reduce the loss more efficiently than the original optimizers at each single step. A natural question is whether these comparison results can be extended to multiple steps. This, however, becomes challenging because, after the first step, the two optimizers explore different regions of the loss landscape, making it easy to construct counterexamples where one method outperforms the other. This is consistent with the *no free lunch* theorems, which state that no single optimizer can dominate another across all possible loss functions (Wolpert & Macready, 1997). Nevertheless, it is reasonable to expect that the advantage observed in a single step would naturally extend to multiple steps for the practical loss functions encountered in deep learning, as evidenced by the experiments presented.

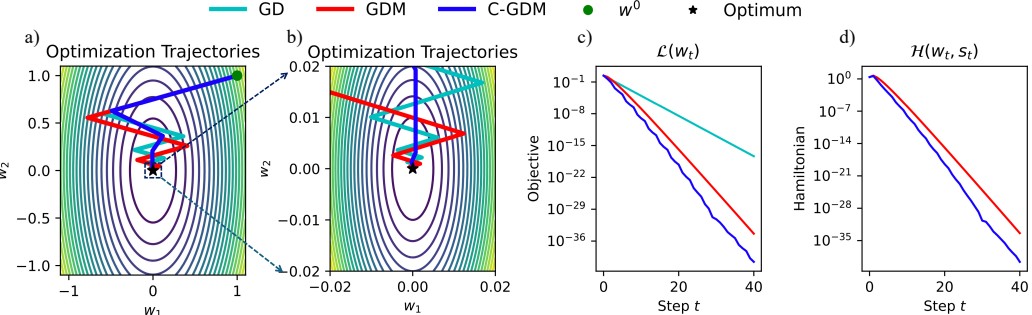

Figure 1: We compare gradient descent with Polyak momentum (GDM) and its element-wise cautious variant (C-GDM), using gradient descent (GD) as a baseline. The step size for GD and the hyperparameters of GDM (including step size and momentum coefficients) are chosen to achieve the optimal convergence rates, which can be analytically derived (see, e.g., Goh (2017)). For cautious optimizers, step sizes $\epsilon$ and momentum coefficients $\beta$ are empirically tuned, as shown in Figure 2. Detailed experimental settings are described in Section 3.1. In **Plot (a)**, we visualize the optimization trajectories of the three methods, starting from the initial point $(1, 1)$ with zero-initialized momentum. Notably, C-GDM converges to the optimum with significantly reduced overshooting and oscillation, **Plot (b)** zooms in on the trajectories from **Plot (a)**, focusing on a smaller region $(0.02 \times 0.02)$ for enhanced clarity. Furthermore, **Plots (c)** and **(d)** show that C-GDM consistently and monotonically decreases both the objective and the Hamiltonian associated with the original GDM, highlighting its superior performance in minimizing these metrics compared to GDM.

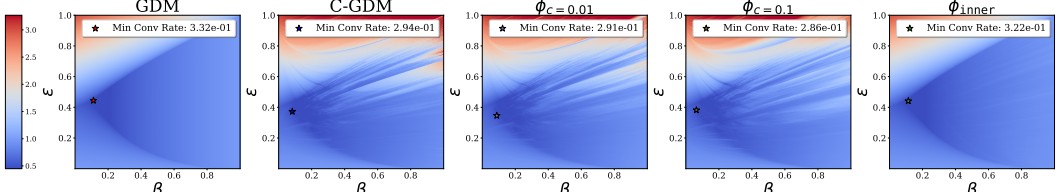

Figure 2: Convergence rate heatmaps for the objective function with condition number $\kappa = 4$ (same setup as in 1). The heatmaps illustrate convergence rates, where values greater than 1 indicate divergence for the corresponding $(\epsilon, \beta)$ configuration. Smaller convergence rates correspond to faster convergence. From left to right, we show heatmaps for $\phi_c$ with $c = 0.01, 0.1$ and $\phi_{\text{inner}}$ as shown in (13). Note that $\phi_0$ represents a constant function, reducing to gradient descent with momentum (GDM), whose optimal convergence rate is given in closed form (Goh, 2017). From the heatmaps, it is clear that all cautious momentum variants $\phi_c$ with $c = 0.01, 0.1$ and $\phi_{\text{inner}}$ demonstrate superior convergence rates compared to GDM.

## 3 EXPERIMENTS

In this section, we evaluate the performance of cautious optimizers compared to their standard counterparts, highlighting the benefits introduced by the cautious masking mechanism. We begin with a 2D toy experiment to provide a visual demonstration of how cautious masking improves optimization. Subsequently, we extend the evaluation to large-scale pretraining tasks for both language and vision models, comparing the performance of standard optimizers and their cautious variants.

### 3.1 2D OPTIMIZATION TOY

We consider a 2D optimization problem with objective $\mathcal{L}(\boldsymbol{w}) = \kappa(w_1)^2 + (w_2)^2$, where $\boldsymbol{w} = (w_1, w_2) \in \mathbb{R}^2$ is the parameter. Obviously, the optimum is at $\boldsymbol{w}^* = (0, 0)$. We set $\kappa = 4$ in our experiments. We apply gradient descent (GD), gradient descent with Polyak momentum (GDM), and cautious gradient descent with momentum (C-GDM) on this toy example, starting from $\boldsymbol{w}_0 = (1, 1)$. Specifically, for GDM, we adopt the conventional momentum update:

$$\boldsymbol{s}_t \leftarrow \beta \boldsymbol{s}_{t-1} + \nabla \mathcal{L}(\boldsymbol{w}_t), \qquad\qquad \boldsymbol{w}_t \leftarrow \boldsymbol{w}_{t-1} - \epsilon \boldsymbol{s}_t,$$

where $\beta \in [0, 1)$ and $\epsilon$ is the learning rate.

**Observation:** On the right of Figure 1, we compare GDM and C-GDM with the same hyperparameters $(\epsilon, \beta)$. We ablate over different combinations of $(\beta, \epsilon) \in \{(0.01, 0.5), (0.01, 0.9), (0.01, 0.99), (0.01, 0.999), (0.1, 0.99), (0.001, 0.99)\}$. Across all settings, C-GDM outperforms GDM, confirming the importance of cautious masking. Given the same $(\epsilon, \beta)$, cautious is always not worse than momentum, and often it gives significant improvement, especially when the choice $(\epsilon, \beta)$ is suboptimal for momentum, meaning that cautious masking makes it more robust. From the left of Figure 1, one can see that GDM, due to the momentum, has fluctuating $\mathcal{L}(\boldsymbol{w}_t)$, while C-GDM ensures that $\mathcal{L}(\boldsymbol{w}_t)$ monotonically decreases. In addition, C-GDM achieves a faster drop in terms of GDM's Hamiltonian.

In Figure 1, we compare GDM and C-GDM, each using their optimal $(\epsilon, \beta)$. For GDM, the optimal values are derived theoretically (e.g., (Goh, 2017)), while for C-GDM, they are obtained through a grid search. Despite using the optimal configuration, GDM exhibits significant overshooting, oscillations, and slower loss convergence. In contrast, C-GDM achieves smoother trajectories, reduced overshooting, and faster convergence, demonstrating its superior stability and efficiency.

We estimate an algorithm's convergence rate as the slope of $\log \mathcal{L}(\boldsymbol{w}_t)$ over time via linear regression and plot heatmaps over $(\epsilon, \beta)$-space, where $\epsilon$ and $\beta$ are the learning rate and momentum. Figure 2 shows that cautious methods achieve lower optimal convergence rates compared to the momentum method ($\frac{\sqrt{\kappa}-1}{\sqrt{\kappa}+1}$, red dot). The heatmaps highlight that all cautious momentum variants outperform GDM in convergence rates.

## 3.2 Pretraining Large Language Models (LLMs)

We begin by investigating the language modeling task using a 100M LLaMA (Touvron et al., 2023) model as the foundational architecture. The models are trained on the C4 (Colossal Clean Crawled Corpus) dataset (Raffel et al., 2020), a large-scale web-crawled text corpus containing billions of tokens. We provide results from the following settings: we take a 100M model and train it with batch size up to 2 million tokens for 50 billion tokens ($25\times$ Chinchilla Optimal). For optimization, we employ AdamW (Loshchilov, 2017) and Lion (Chen et al., 2023c), two popular optimizers in modern language modeling, as baselines. These are compared with their cautious counterparts, which we term Cautious AdamW (C-AdamW) and Cautious Lion (C-Lion).

| lr | 1e-4 | 3e-4 | 1e-3 | 3e-3 | 1e-2 | 2e-2 | 3e-2 | 1e-1 |
|---|---|---|---|---|---|---|---|---|
| AdamW | 85.050 | 24.384 | 19.249 | 19.007 | **18.965** | 19.609 | ** | ** |
| C-AdamW | – | – | 19.065 | 18.771 | **18.684** | 18.821 | – | – |

| | lr | 3e-5 | 1e-4 | 3e-4 | 6e-4 | 1e-3 | 1e-2 |
|---|---|---|---|---|---|---|---|
| | Lion | ** | 28.250 | **21.401** | 21.937 | ** | ** |
| | C-Lion | – | 21.354 | **19.795** | 20.403 | 20.977 | ** |

Table 1: We report final evaluation perplex, the lower the better. "**" are runs that did not converge due to either too large or too small learning rates. "–" stands for runs we skip due to lack of baseline comparison. Each model is trained with batch size up to **2 million** tokens for **50 billion** tokens in total ($25 \times$ Chinchilla Optimal(Hoffmann et al., 2022)). We use $\beta_1 = 0.9$, $\beta_2 = 0.95$ and weight decay 0.1 on AdamW; Sequence lengths of all models are 1024. For Lion experiments, we follow the recommendation from the (Chen et al., 2023a) and use $\beta_1 = 0.95$ and $\beta_2 = 0.98$. For scheduler, we use CosineAnnealing with warmup and the learning rate is decayed to 10% of the initial learning rate. Gradient accumulation is set to 8 to increase the global batch size.

**Observation:** As shown in Table 1, Cautious Optimizers demonstrate consistent improvements in both evaluation perplexity and sample efficiency. Table 1 shows the cautious modification is robust across learning rates and Cautious doesn't change the optimality of hyperparameter search done on the base optimizers. Surprisingly, Cautious can also tolerate a higher learning rate in the Lion experiments and achieve stable training even when the baseline diverges.

To further confirm our finding, we also include a scaling experiment with C-AdamW on FineWeb-Edu (Penedo et al., 2024), a more recent and higher quality web-scale text dataset. After rigorous and thorough hyperparameter search, we found that C-AdamW is consistently outperforming baseline AdamW. In table 2, we report optimal results for each scale.

Table 2: Perplexity comparison between AdamW and C-AdamW across different model scales at $1\times$ Chinchilla (Hoffmann et al., 2022), hyperparameters are extensively searched with coordinate descent over a discrete grid. Details can be found in appendix C

| Scale | AdamW | C-AdamW | Improvement (%) |
|-------|-------|---------|-----------------|
| 130M | 27.39 | **27.30** | 0.33 |
| 300M | 18.30 | **18.28** | 0.10 |
| 520M | 15.07 | **14.92** | 1.00 |
| 1.2B | 11.36 | **11.32** | 0.32 |

Furthermore, we perform downstream evaluation on the produced 1.2B checkpoints with $1\times$ Chinchilla ($20\times$ tpp) on 7 downstream tasks, where the checkpoint trained by cautious optimizer wins in 5 of them (MMLU, OpenBookQA, Arc Easy as well as HellaSwag and Arc Challenge).

| Task / Group | Metric | AdamW | C-AdamW |
|--------------|--------|-------|---------|
| Arc Easy (Clark et al., 2018) | acc | $0.6082 \pm 0.0100$ | $\mathbf{0.6090 \pm 0.0100}$ |
| Arc Challenge (Clark et al., 2018) | acc_norm | $0.2875 \pm 0.0132$ | $\mathbf{0.2978 \pm 0.0134}$ |
| Hellaswag (Zellers et al., 2019) | acc_norm | $0.4169 \pm 0.0049$ | $\mathbf{0.4193 \pm 0.0049}$ |
| Lambada OpenAI (Radford et al., 2019) | acc | $\mathbf{0.3311 \pm 0.0066}$ | $0.3229 \pm 0.0065$ |
| OpenBookQA (Mihaylov et al., 2018) | acc | $0.2340 \pm 0.0190$ | $\mathbf{0.2360 \pm 0.0190}$ |
| PIQA (Bisk et al., 2020) | acc_norm | $\mathbf{0.6774 \pm 0.0109}$ | $0.6768 \pm 0.0109$ |
| MMLU (Hendrycks et al., 2021) | acc | $0.2529 \pm 0.0037$ | $\mathbf{0.2535 \pm 0.0037}$ |

Table 3: Comparison of benchmark results between C-AdamW and AdamW across multiple tasks and MMLU groups. All evaluations are done with LM Eval-Harness(Gao et al., 2024). Bold indicates the better score.

## 3.3 IMAGE CLASSIFICATION

We also include a classic classification task on Mini-ImageNet on ViT (Dosovitskiy, 2020) with two additional optimizers MARS (Yuan et al., 2024) and LaProp (Ziyin et al., 2020), both are momentum-based optimizers and their cautious variants perform consistently better as shown in the table 4.

| Method | Eval_Top1 |
|--------|-----------|
| C-AdamW | **73.52** |
| AdamW | 72.11 |
| C-LaProp | **73.92** |
| LaProp (Ziyin et al., 2020) | 71.73 |
| C-MARS | **74.91** |
| MARS (Yuan et al., 2024) | 74.06 |

Table 4: Top-1 evaluation accuracy on Mini-ImageNet, the higher the better. We can see that the cautious variant is better across base optimizer options. Hyperparameters can be found in appendix D

## 4 RELATED WORK

We provide a brief overview of existing efforts on designing Adam-like optimizers, and the related works on Hamiltonian dynamics.

**Adam and Its Variants** A plethora of Adam variants have been developed to address different aspects of optimization challenges (Kingma, 2014; Loshchilov & Hutter, 2017). AdamW (Loshchilov & Hutter, 2017) introduced a key improvement by decoupling weight decay from optimization steps, restoring the original formulation of weight decay regularization. NAdam (Dozat, 2016) incorporated Nesterov updates into Adam, while AdaBelief (Zhuang et al., 2020) refined the second momentum

$v_t$ to track the EMA of $(g_t - m_t)^2$, improving generalization. Adan (Xie et al., 2024) added an extra momentum term for better training performance, albeit at the cost of additional memory usage. More recently, ADOPT (Taniguchi et al., 2024) innovated by folding normalized updates into first-order momentum updates. From an efficiency perspective, several approaches target memory cost reduction. AdaFactor (Shazeer & Stern, 2018) factorizes second-order statistics into a row-column outer product, enabling sub-linear memory usage. K-Fac (Martens & Grosse, 2015) approximates the Fisher information matrix with a Kronecker-factored representation, supporting sublinear natural gradient updates. Techniques like fused gradient computation (Lv et al., 2023) further minimize memory costs.(Wang et al., 2024) is a concurrent work that focus on continuous learning setting. In contrast to these approaches, our proposed C-AdamW introduces a single-line modification to the widely used AdamW optimizer. This modification not only retains the simplicity and efficiency of AdamW but also eliminates the need for hyperparameter tuning, as the default parameters of AdamW suffice. Furthermore, while many of the aforementioned methods focus on optimizing or extending specific aspects of the Adam algorithm, C-AdamW is more generalit seamlessly integrates with all momentum-based optimizers, offering a general solution with minimal implementation effort.

**Hamiltonian Dynamics**   Hamiltonian dynamics, rooted in classical mechanics, provides a powerful mathematical framework for analyzing the motion of systems in continuous spaces. This perspective has gained traction in optimization, where the introduction of Hamiltonian principles sheds light on momentum-based algorithms (Sutskever et al., 2013; Nesterov, 1983; Nguyen et al., 2024; Anonymous, 2024). Unlike Gradient Descent (GD), which ensures a monotonic decrease in the objective function, momentum-based methods often follow non-monotonic trajectories, posing unique analytical challenges (Jin et al., 2018). To address this, researchers have developed multiple Lyapunov functions for convex settings (Krichene et al., 2015; Wilson et al., 2016), providing a structured approach to analyze convergence. (Sutskever et al., 2013) offered a physical interpretation of momentum in optimization, linking it to the dynamics described by Hamiltonian mechanics, and demonstrated how these principles underpin classical methods like those of Nesterov and Polyak (Nesterov, 1983). Furthermore, Hamiltonian dynamics have been instrumental in deriving convergence rates for accelerated methods (Jin et al., 2018) and, more recently, for advanced optimizers like Lion (Chen et al., 2023a) and its distributed variant (Liu et al., 2024). In a related vein, (Maddison et al., 2018) explored optimization methods from the perspective of continuous-time ODEs, emphasizing their Hamiltonian structure. Mirror Descent, a related framework, has been shown to maintain efficiency estimates with a mild dependence on the dimensionality of decision variables, making it particularly suitable for large-scale optimization problems (Tzen et al., 2023; Krichene et al., 2015). These advancements highlight the versatility and depth of Hamiltonian formalism in bridging the gap between optimization theory and practical algorithm design.

## 5   CONCLUSION

In summary, we introduce Cautious Optimizers, an enhancement for momentum-based optimizers that can be implemented with a single line of code. Our theoretical analysis demonstrates that Cautious Optimizers not only preserve the convergence guarantees of the base optimizers but also accelerate the reduction of the loss function. Empirically, it delivers faster LLM pretraining and better accuracy on image classification. Finally, we suggest a few promising future directions: (1) Apply cautious optimizers to more settings such as reinforcement learning and continuous learning; (2) masking in the eigenspace rather than the parameter space; (3) rigorous analyses of how cautious optimizers strictly improve the convergence rate (empirically shown in 1).

## 6   ACKNOWLEDGEMENT

This work was supported by the Amazon AI PhD Fellowship, and in part by the Institute for Foundations of Machine Learning (IFML) and the Office of Naval Research (ONR) under Grant No. N00014-25-1-2354.

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

# A APPENDIX

The appendix is organized into the following key components:

- **Cautious Hamiltonian Descent**: We provide conditions 2.1 on the choice of function $\phi$ ensure that the system decreases *both* $\mathcal{H}$ and $\mathcal{L}$ simultaneously and Corollary 2.2 asserts that with a positive definite norm and differentiable Hamiltonian, bounded solutions of the discussed systems converge to a stationary point of the Hamiltonian.

- **Applications to Optimizers**: In section A.3, the cautious framework is applied to derive cautious variants of popular optimizers, including Adam, Signed Momentum, and Lion, highlighting their theoretical properties and practical advantages.

- **Discrete-Time Analysis**: A rigorous analysis connects the continuous-time dynamics to discrete-time updates, providing convergence guarantees and bounds on loss reduction for cautious optimizers.

- **Pseudocode**: We present the implementation details for cautious optimizers, focusing on the cautious Lion optimizer as a representative example.

- **Experimental Details**: Comprehensive details on the experimental setup, hyperparameters, and hardware configurations are provided, demonstrating the effectiveness of cautious optimizers in accelerating convergence and improving performance on large-scale tasks such as language modeling and masked autoencoder pretraining.

## A.1 HAMILTONIAN OF COMMON OPTIMIZERS

We introduce the Hamiltonian functions of the common optimizers.

**Example A.1.** *Adam (Kingma, 2014) yields the following continuous-time form and Hamiltonian,*

$$\frac{\mathrm{d}}{\mathrm{d}t}\boldsymbol{v}w_t = -\frac{\boldsymbol{v}m_t}{\sqrt{\boldsymbol{v}v_t}+e}, \qquad\qquad \frac{\mathrm{d}}{\mathrm{d}t}\boldsymbol{v}m_t = a(\nabla\mathcal{L}(\boldsymbol{v}w_t) - \boldsymbol{v}m_t),$$

$$\frac{\mathrm{d}}{\mathrm{d}t}\boldsymbol{v}v_t = b(\nabla\mathcal{L}(\boldsymbol{v}w_t)^{\odot 2} - \boldsymbol{v}v_t),$$

$$\text{with}\quad \mathcal{H}(\boldsymbol{v}w, \boldsymbol{v}m, \boldsymbol{v}v) = \mathcal{L}(\boldsymbol{v}w) + \frac{1}{2a}\langle\frac{\boldsymbol{v}m}{\sqrt{\boldsymbol{v}v}+e}, \; \boldsymbol{v}m\mathrm{a}.$$

*We can show that $\frac{\mathrm{d}}{\mathrm{d}t}\mathcal{H}(\boldsymbol{v}w_t, \boldsymbol{v}m_t, \boldsymbol{v}v_t) \leq 0$ when $a \geq b/4$.*

**Example A.2.** *The Lion-$\mathcal{K}$ optimizer (Chen et al., 2023b;a) (without weight decay) can be written into*

$$\frac{\mathrm{d}}{\mathrm{d}t}\boldsymbol{v}w_t = \nabla\mathcal{K}((1-b)\boldsymbol{v}m_t - b\nabla\mathcal{L}(\boldsymbol{v}w_t)),$$
$$\frac{\mathrm{d}}{\mathrm{d}t}\boldsymbol{v}m_t = -a(\nabla\mathcal{L}(\boldsymbol{v}w_t) + \boldsymbol{v}m_t)$$

*where $a \geq 0$, $b \in [0, 1]$ and $\mathcal{K}(vx)$ is any convex function that attains the minimum at $vx = 0$. One of its Hamiltonians that yields the Hamiltonian+descent structure (Eq (13) in Chen et al. (Chen et al., 2023a)) is*

$$\mathcal{H}(\boldsymbol{v}w, \boldsymbol{v}m) = a\mathcal{L}(\boldsymbol{v}w) + \frac{1}{1-b}\mathcal{K}((1-b)\boldsymbol{v}m).$$

*See Chen et al. (2023a) for other Hamiltonian functions. Lion-$\mathcal{K}$ includes a large family algorithms as special cases, including Polyka momentum, Nesterov momentum, signed momentum, mirror descent, Frank-Wolfe, etc.*

## A.2 Cautious Dynamics

We establish conditions in Theorem 2.1 for the function $\phi$ that ensure simultaneous decreases in both $\mathcal{H}$ and $\mathcal{L}$. Corollary 2.2 further asserts that, with a positive definite norm and differentiable Hamiltonian, bounded solutions of the systems converge to a stationary point of the Hamiltonian.

**Theorem 2.1.** Following the dynamics in (5) in $\mathbb{R}^d$, we have

$$\frac{\mathrm{d}}{\mathrm{d}t}\mathcal{H}(\overline{\boldsymbol{w}}_t, \overline{\boldsymbol{s}}_t) = (\overline{\boldsymbol{x}}_t^\top (\mathbf{1} - \phi(\overline{\boldsymbol{x}}_t)) - \Delta_{\mathcal{H}_t}(\overline{\boldsymbol{w}}_t, \overline{\boldsymbol{s}}_t),$$

and

$$\begin{aligned}
\frac{\mathrm{d}}{\mathrm{d}t}\mathcal{L}(\overline{\boldsymbol{w}}_t) &= -\overline{\boldsymbol{x}}_t^\top \phi(\overline{\boldsymbol{x}}_t) - \|\nabla\mathcal{L}(\overline{\boldsymbol{w}}_t)\|_{\Phi_t}^2 \\
&= (\overline{\boldsymbol{x}}_t^\top (\mathbf{1} - \phi(\overline{\boldsymbol{x}}_t)) - \Delta_{\mathcal{L}_t}(\overline{\boldsymbol{w}}_t, \overline{\boldsymbol{s}}_t),
\end{aligned}$$

Here, $\Delta_{\mathcal{H}_t}(\overline{\boldsymbol{w}}_t, \overline{\boldsymbol{s}}_t)$ and $\Delta_{\mathcal{L}_t}(\overline{\boldsymbol{w}}_t, \overline{\boldsymbol{s}}_t)$, as defined in (3) and (4), respectively, represent the decreasing rates of $\mathcal{H}$ and $\mathcal{L}$ in accordance with the original system (2). Hence:

- If $\boldsymbol{x}^\top (\mathbf{1} - \phi(\boldsymbol{x})) \leq 0$ for any $\boldsymbol{x} \in \mathbb{R}^d$, then both $\mathcal{H}$ and $\mathcal{L}$ decreases faster than the original system:

$$\begin{aligned}
\frac{\mathrm{d}}{\mathrm{d}t}\mathcal{H}(\overline{\boldsymbol{w}}_t, \overline{\boldsymbol{s}}_t) &\leq -\Delta_{\mathcal{H}_t}(\overline{\boldsymbol{w}}_t, \overline{\boldsymbol{s}}_t) \leq 0, \\
\frac{\mathrm{d}}{\mathrm{d}t}\mathcal{L}(\overline{\boldsymbol{w}}_t) &\leq -\Delta_{\mathcal{L}_t}(\overline{\boldsymbol{w}}_t, \overline{\boldsymbol{s}}_t).
\end{aligned}$$

- If $\boldsymbol{x}^\top \phi(\boldsymbol{x}) \geq 0$ for any $\boldsymbol{x} \in \mathbb{R}^d$, then $\mathcal{L}$ decreases monotonically,

$$\frac{\mathrm{d}}{\mathrm{d}t}\mathcal{L}(\overline{\boldsymbol{w}}_t) \leq 0.$$

*Proof.* For simplicity, we write $\overline{\boldsymbol{x}}_t = \nabla\mathcal{L}(\overline{\boldsymbol{w}}_t) \circ \nabla\mathcal{K}(\overline{\boldsymbol{s}}_t)$.

Recall the definition of $\Delta_{\mathcal{H}_t}(\overline{\boldsymbol{w}}_t, \overline{\boldsymbol{s}}_t)$ and $\Delta_{\mathcal{L}_t}(\overline{\boldsymbol{w}}_t, \overline{\boldsymbol{s}}_t)$:

$$\begin{aligned}
\Delta_{\mathcal{H}}(\overline{\boldsymbol{w}}_t, \overline{\boldsymbol{s}}_t) &\coloneqq \|\nabla\mathcal{L}(\overline{\boldsymbol{w}}_t)\|_{\Phi_t}^2 + \|\nabla\mathcal{K}(\overline{\boldsymbol{s}}_t)\|_{\Psi_t}^2 \\
\Delta_{\mathcal{L}}(\overline{\boldsymbol{w}}_t, \overline{\boldsymbol{s}}_t) &\coloneqq \nabla\mathcal{L}(\overline{\boldsymbol{w}}_t)^\top \nabla\mathcal{K}(\overline{\boldsymbol{s}}_t) + \|\nabla\mathcal{L}(\overline{\boldsymbol{w}}_t)\|_{\Phi_t}^2.
\end{aligned}$$

Following the dynamics in (5), let us see the derivation of $\mathcal{H}$ w.r.t. $t$:

$$\begin{aligned}
&\frac{\mathrm{d}}{\mathrm{d}t}\mathcal{H}(\overline{\boldsymbol{w}}_t, \overline{\boldsymbol{s}}_t) \\
&= \nabla\mathcal{L}(\overline{\boldsymbol{w}}_t)^\top \dot{\overline{\boldsymbol{w}}}_t + \nabla\mathcal{K}(\overline{\boldsymbol{s}}_t)^\top \dot{\overline{\boldsymbol{s}}}_t \\
&= \nabla\mathcal{L}(\overline{\boldsymbol{w}}_t)^\top (-\nabla\mathcal{K}(\overline{\boldsymbol{s}}_t) \circ \phi(\overline{\boldsymbol{x}}_t) - \Phi_t(\nabla\mathcal{L}(\overline{\boldsymbol{w}}_t))) + \nabla\mathcal{K}(\overline{\boldsymbol{s}}_t)^\top (\nabla\mathcal{L}(\overline{\boldsymbol{w}}_t) - \Psi_t(\nabla\mathcal{K}(\overline{\boldsymbol{s}}_t))) \\
&= \nabla\mathcal{L}(\overline{\boldsymbol{w}}_t)^\top (\nabla\mathcal{K}(\overline{\boldsymbol{s}}_t) \circ (\mathbf{1} - \phi(\overline{\boldsymbol{x}}_t))) - \nabla\mathcal{L}(\overline{\boldsymbol{w}}_t)^\top \Phi_t(\nabla\mathcal{L}(\overline{\boldsymbol{w}}_t)) - \nabla\mathcal{K}(\overline{\boldsymbol{s}}_t)^\top \Psi_t(\nabla\mathcal{K}(\overline{\boldsymbol{s}}_t)) \\
&= \mathbf{1}^\top ((\nabla\mathcal{L}(\overline{\boldsymbol{w}}_t) \circ \nabla\mathcal{K}(\overline{\boldsymbol{s}}_t)) \circ (\mathbf{1} - \phi(\nabla\mathcal{L}(\overline{\boldsymbol{w}}_t) \circ \nabla\mathcal{K}(\overline{\boldsymbol{s}}_t)))) - \|\nabla\mathcal{L}(\overline{\boldsymbol{w}}_t)\|_{\Phi}^2 - \|\nabla\mathcal{K}(\overline{\boldsymbol{s}}_t)\|_{\Psi_t}^2 \\
&= \overline{\boldsymbol{x}}_t^\top (\mathbf{1} - \phi(\overline{\boldsymbol{x}}_t)) - \Delta_{\mathcal{H}}(\overline{\boldsymbol{w}}_t, \overline{\boldsymbol{s}}_t). \quad (9)
\end{aligned}$$

Given the fact that $\phi$ is an element-wise operator, it is noteworthy that if $x \cdot (1 - \phi(x)) \leq 0$, then the first term in (9) $\mathbf{1}^\top \left( (\nabla \mathcal{L}(\overline{w}) \circ \nabla K(\overline{s})) \circ (\mathbf{1} - \phi(\nabla \mathcal{L}(\overline{w}) \circ \nabla K(\overline{s}))) \right) \leq 0$ since we see $\nabla \mathcal{L}(\overline{w}) \circ \nabla \mathcal{K}(\overline{s})$ as $\overline{x}$.

Next, let us look into the derivative of $\mathcal{L}(\overline{w}_t)$ w.r.t. $t$:

$$
\begin{aligned}
\frac{\mathrm{d}}{\mathrm{d}t} \mathcal{L}(\overline{w}_t) &= \nabla \mathcal{L}(\overline{w}_t)^\top \left( -\nabla \mathcal{K}(\overline{s}_t) \circ \phi(\nabla \mathcal{L}(\overline{w}_t) \circ \nabla \mathcal{K}(\overline{s}_t)) - \Phi_t(\nabla \mathcal{L}(\overline{w}_t)) \right) \\
&= -\nabla \mathcal{L}(\overline{w}_t)^\top \left( \nabla \mathcal{K}(\overline{s}_t) \circ \phi(\nabla \mathcal{L}(\overline{w}_t) \circ \nabla \mathcal{K}(\overline{s}_t)) \right) - \nabla \mathcal{L}(\overline{w}_t)^\top \Phi_t(\nabla \mathcal{L}(\overline{w}_t)) \\
&= -\overline{x}_t^\top \phi(\overline{x}_t) - \|\nabla \mathcal{L}(\overline{w}_t)\|_{\Phi_t}^2 \qquad\qquad (10) \\
&= (\overline{x}_t^\top (\mathbf{1} - \phi(\overline{x}_t)) - \Delta_{\mathcal{L}_t}(\overline{w}_t, \overline{s}_t). \qquad\qquad (11)
\end{aligned}
$$

It is evident that if $x^\top \phi(x) \geq 0$ for any $x \in \mathbb{R}^d$, then the first term in (10), $\overline{x}_t^\top \phi(\overline{x}_t)$, satisfies $\overline{x}_t^\top \phi(\overline{x}_t) \geq 0$. Consequently, $\frac{d}{dt} \mathcal{L}(\overline{w}_t) \leq 0$. This condition holds when $x^\top \phi(x) \geq 0$ for all $x \in \mathbb{R}^d$.

It is noteworthy that if $x^\top (\mathbf{1} - \phi(x)) \leq 0$ for any $x \in \mathbb{R}^d$, then the first term in (11), $\overline{x}_t^\top (\mathbf{1} - \phi(\overline{x}_t)) \leq 0$, holds. Consequently, we have $\frac{d}{dt} \mathcal{L}(\overline{w}_t) \leq -\Delta_{\mathcal{L}_t}(\overline{w}_t, \overline{s}_t)$.

$\square$

**Corollary 2.2.** Assume that the norm $\| \cdot \|_\Psi^2$ is positive definite, $\Psi(0) = 0$, and that $\mathcal{H}(w, s) = \mathcal{L}(w) + \mathcal{K}(s)$ is differentiable. Then, the bounded solutions of the original system (2) converge to a stationary point of $\mathcal{H}(w, s)$. Similarly, the bounded solutions of (5) also converge to a stationary point of $\mathcal{H}(w, s)$.

*Proof.* First, Let us look at system (2), we use LaSalle's invariance principle to find the conditions that the accumulation points (positive limit points)$(w^*, s^*)$ satisfy:

$$
\|\nabla \mathcal{L}(w^*)\|_{\Phi_t}^2 = \|\nabla \mathcal{K}(s^*)\|_{\Psi_t}^2 = 0.
$$

By the assumption that $\| \cdot \|_{\Psi_t}^2$ is positive definite and $\Psi_t(0) = 0$, we have $\nabla \mathcal{K}(s^*) = 0$.

For positive limit points of systems (2), if $\nabla \mathcal{L}(w^*) \neq 0$, then the point $(w^*, s^*)$ is not a positive limit point since

$$
\dot{s}^* = \nabla \mathcal{L}(w^*) - \Psi_t(\nabla \mathcal{K}(s^*)) = \nabla \mathcal{L}(w^*) \neq 0.
$$

Thus, $\nabla \mathcal{L}(w^*) = 0$. Together with $\nabla \mathcal{K}(s^*) = 0$, we conclude that $(w^*, s^*)$ is a stationary point of $\mathcal{H}(w, s)$.

For system (5), the proof follows the same reasoning.

$\square$

### A.3 EXAMPLES OF CAUTIOUS DYNAMICS

In this section, we instantiate results on Adam, Signed Momentum, and Lion, along with their cautious variants. While the standard methods are widely used, our focus is on introducing and analyzing the cautious versions of these algorithms to better understand their dynamics and stability.

#### A.3.1 CAUTIOUS ADAM

**Adam:** We begin by recalling the continuous-time dynamics of the Adam optimizer:

$$
\begin{aligned}
\frac{\mathrm{d}}{\mathrm{d}t} w_t &= -\frac{m_t}{\sqrt{v_t} + \epsilon}, \\
\frac{d}{dt} m_t &= \beta_1 \cdot (\nabla \mathcal{L}(w_t) - m_t), \\
\frac{d}{dt} v_t &= \beta_2 \cdot (\nabla \mathcal{L}(w_t)^{\odot 2} - v_t),
\end{aligned}
$$

where the associated Hamiltonian is given by:

$$\mathcal{H}(\boldsymbol{w}_t, \boldsymbol{m}_t, \boldsymbol{v}_t) = \mathcal{L}(\boldsymbol{w}_t) + \frac{1}{2\beta_1} \left\langle \frac{\boldsymbol{m}_t}{\sqrt{\boldsymbol{v}_t} + \epsilon}, \boldsymbol{m}_t \right\rangle. \tag{12}$$

The time evolution of the Hamiltonian can be derived as:

$$\frac{d\mathcal{H}(\boldsymbol{w}_t, \boldsymbol{m}_t, \boldsymbol{v}_t)}{dt} = -\frac{\beta_2}{4\beta_1} \left\langle \frac{\boldsymbol{m}_t^{\odot 2}}{\boldsymbol{v}_t^{3/2} + \epsilon}, \nabla\mathcal{L}(\boldsymbol{w}_t)^{\odot 2} \right\rangle - \left(1 - \frac{\beta_2}{4\beta_1}\right) \left\langle \frac{\boldsymbol{m}_t}{\sqrt{\boldsymbol{v}_t} + \epsilon}, \boldsymbol{m}_t \right\rangle.$$

For stability, we require $\frac{d\mathcal{H}}{dt} \leq 0$, which leads to $\beta_1 \geq \frac{\beta_2}{4}$.

**Cautious Adam:** We now introduce a cautious variant of Adam. In this case, the update for $\boldsymbol{w}_t$ is modified by introducing an indicator function that enforces alignment between the gradient and the momentum:

$$\frac{\mathrm{d}}{\mathrm{d}t}\boldsymbol{w}_t = -\frac{\mathbb{I}(\nabla\mathcal{L}(\boldsymbol{w}_t) \circ \boldsymbol{m}_t > 0) \circ \boldsymbol{m}_t}{\sqrt{\boldsymbol{v}_t} + \epsilon}.$$

The dynamics for $\boldsymbol{m}_t$ and $\boldsymbol{v}_t$ remain the same. It is straightforward to verify that the loss function $\mathcal{L}(\boldsymbol{w}_t)$ serves as a Lyapunov function:

$$\frac{\mathrm{d}}{\mathrm{d}t}\mathcal{L}(\boldsymbol{w}_t) = -\nabla\mathcal{L}(\boldsymbol{w}_t)^{\top} \frac{\mathbb{I}(\nabla\mathcal{L}(\boldsymbol{w}_t) \circ \boldsymbol{m}_t > 0) \circ \boldsymbol{m}_t}{\sqrt{\boldsymbol{v}_t} + \epsilon} \leq 0.$$

Meanwhile, by Theorem 2.1, $\mathcal{H}(\boldsymbol{w}_t, \boldsymbol{m}_t, \boldsymbol{v}_t) = \mathcal{L}(\boldsymbol{w}_t) + \frac{1}{2\beta_1} \left\langle \frac{\boldsymbol{m}_t}{\sqrt{\boldsymbol{v}_t} + \epsilon}, \boldsymbol{m}_t \right\rangle$ is also monotonically decreasing. Therefore, $\mathcal{H}$ remains a valid Hamiltonian for Cautious Adam.

### A.3.2 CAUTIOUS SIGNED (POLYAK) MOMENTUM

**Signed Momentum:** The update rule for Signed Momentum is:

$$\dot{\boldsymbol{w}}_t = -\mathrm{sign}(\boldsymbol{m}_t),$$
$$\dot{\boldsymbol{m}}_t = \nabla\mathcal{L}(\boldsymbol{w}_t) - \boldsymbol{m}_t.$$

The Hamiltonian for this system is:

$$\mathcal{H}(\boldsymbol{w}_t, \boldsymbol{m}_t) = \mathcal{L}(\boldsymbol{w}_t) + \|\boldsymbol{m}_t\|_1,$$

with time evolution:

$$\frac{d\mathcal{H}}{dt} = -\|\boldsymbol{m}_t\|_1.$$

**Cautious Signed Momentum:** The cautious variant modifies the update rule for $\boldsymbol{w}_t$ by introducing an indicator:

$$\dot{\boldsymbol{w}}_t = -\mathrm{sign}(\boldsymbol{m}_t) \odot \mathbb{I}(\nabla\mathcal{L}(\boldsymbol{w}_t) \circ \boldsymbol{m}_t > 0).$$

The Lyapunov function remains the loss $\mathcal{L}(\boldsymbol{w}_t)$, with:

$$\frac{d\mathcal{L}(\boldsymbol{w}_t)}{dt} = -\|\nabla\mathcal{L}(\boldsymbol{w}_t) \odot \mathbb{I}(\nabla\mathcal{L}(\boldsymbol{w}_t) \circ \boldsymbol{m}_t > 0)\|_1.$$

By Theorem 2.1, $\mathcal{H}(\boldsymbol{w}_t, \boldsymbol{m}_t) = \mathcal{L}(\boldsymbol{w}_t) + \|\boldsymbol{m}_t\|_1$ is also monotonically decreasing. Therefore, $\mathcal{H}$ remains a valid Hamiltonian for Cautious Signed Momentum.

### A.3.3 CAUTIOUS LION

**Lion:** The dynamics of Lion Chen et al. (2023a) are given by:

$$\dot{\boldsymbol{m}}_t = \alpha\nabla\mathcal{L}(\boldsymbol{w}_t) - \gamma\boldsymbol{m}_t,$$
$$\dot{\boldsymbol{w}}_t = -\mathrm{sign}(\tilde{\boldsymbol{m}}_t),$$

where $\tilde{\boldsymbol{m}}_t = \boldsymbol{m}_t - \varepsilon(\alpha\nabla\mathcal{L}(\boldsymbol{w}_t) + \gamma\boldsymbol{m}_t)$. The associated Hamiltonian is:

$$\mathcal{H}(\boldsymbol{w}_t, \boldsymbol{m}_t) = \alpha\mathcal{L}(\boldsymbol{w}_t) + (1 - \varepsilon\gamma)\|\boldsymbol{m}_t\|_1.$$

The time derivative is:

$$\dot{\mathcal{H}}(\boldsymbol{w}_t, \boldsymbol{m}_t) = -(1 - \varepsilon\gamma)\|\tilde{\boldsymbol{m}}_t\|_1 - \gamma\|\boldsymbol{m}_t\|_1.$$

**Cautious Lion:** In the cautious version, the update for $w_t$ is modified:

$$\dot{w}_t = -\text{sign}(\tilde{m}_t) \odot \mathbb{I}(\nabla\mathcal{L}(w_t) \circ \tilde{m}_t > 0).$$

By Theorem 2.1, $\mathcal{H}(w_t, m_t) = \alpha\mathcal{L}(w_t) + (1 - \varepsilon\gamma)\|m_t\|_1$ is also monotonically decreasing. Therefore, $\mathcal{H}$ remains a valid Hamiltonian for Cautious Lion.

Meanwhile, the Lyapunov function remains $\mathcal{L}(w_t)$, with:

$$\frac{d\mathcal{L}(w_t)}{dt} = -\|\nabla\mathcal{L}(w_t) \odot \mathbb{I}(\nabla\mathcal{L}(w_t) \circ \tilde{m}_t > 0)\|_1.$$

## A.4 DISCRETE TIME ANALYSIS

We analyze the discrete time case, demonstrating that cautious optimizers are at least as good as the original optimizers under mild conditions.

**Theorem 2.3.** Consider (7) and (8) with the loss function $\mathcal{L}(\cdot)$ $\mu$-smooth. Assume the element-wise operator $\phi$ satisfies

$$\Delta(vx) := -vx^\top(1 - \phi(vx)) \geq 0.$$

starting from $(w_t, s_t) = (\overline{w}_t, \overline{s}_t)$, we have

$$\mathcal{L}(\overline{w}_{t+1}) \leq \mathcal{L}(w_{t+1}),$$

which holds for step size $\epsilon_t \leq \frac{2\Delta(\overline{vu}_t \circ \overline{vg}_t)}{\mu\|\overline{vr}_t\|(2\cdot\|\overline{u}_t\|+\|\overline{vr}_t\|)}$, where $\overline{vr}_t = \overline{u}_t \circ (1 - \overline{v\phi}_t)$ and $\overline{vg}_t = \nabla\mathcal{L}(\overline{w}_t)$.

*Proof.* First, we calculate the difference between $w_{t+1}$ and $\overline{w}_{t+1}$, then we use $\mu$-smooth condition to bound the difference between $\mathcal{L}(w_{t+1})$ and $\mathcal{L}(\overline{w}_{t+1})$.

$$\overline{w}_{t+1} - w_{t+1} = -\epsilon_t(\overline{u}_t \circ \overline{v\phi}_t - u_t) = -\epsilon_t(\overline{u}_t \circ \overline{v\phi}_t - \overline{u}_t) = \epsilon_t\overline{vr}_t.$$

Let us use $\mu$-smooth condition to bound the $\mathcal{L}$ difference

$\mathcal{L}(\overline{w}_{t+1}) - \mathcal{L}(w_{t+1})$

$$\leq \epsilon_t\nabla\mathcal{L}(w_{t+1})^\top\overline{vr}_t + \frac{\mu}{2}\epsilon_t^2\|\overline{vr}_t\|^2$$

$$= \epsilon_t\left(\nabla\mathcal{L}(w_t) + \nabla\mathcal{L}(w_{t+1}) - \nabla\mathcal{L}(w_t)\right)^\top\overline{vr}_t + \frac{\mu}{2}\epsilon_t^2\|\overline{vr}_t\|^2$$

$$= \epsilon_t\nabla\mathcal{L}(w_t)^\top\overline{vr}_t + \epsilon_t\left(\nabla\mathcal{L}(w_{t+1}) - \nabla\mathcal{L}(w_t)\right)^\top\overline{vr}_t + \frac{\mu}{2}\epsilon_t^2\|\overline{vr}_t\|^2$$

$$\leq \epsilon_t\nabla\mathcal{L}(w_t)^\top\overline{vr}_t + \epsilon_t\|\nabla\mathcal{L}(w_{t+1}) - \nabla\mathcal{L}(w_t)\|\|\overline{vr}_t\| + \frac{\mu}{2}\epsilon_t^2\|\overline{vr}_t\|^2 \quad \text{//Cauchy Schwarz inequality}$$

$$= \epsilon_t\overline{vg}_t^\top\overline{vr}_t + \epsilon_t^2\mu\|\overline{vu}_t\|\|\overline{vr}_t\| + \frac{\mu}{2}\epsilon_t^2\|\overline{vr}_t\|^2 \quad \text{//}\overline{vg}_t^\top\overline{vr}_t \leq 0.$$

$$= -\epsilon_t\Delta(\overline{u}_t \circ \overline{vg}_t) + \epsilon_t^2\mu\|\overline{vu}_t\|\|\overline{vr}_t\| + \frac{\mu}{2}\epsilon_t^2\|\overline{vr}_t\|^2$$

$$\leq 0. \quad \text{//By the choice of } \epsilon_t.$$

$\square$

**Theorem A.3.** *Assume $\mathcal{L}(w)$ is $\mu$-smooth and differentiable, and the element-wise operator $\phi$ satisfies $x \cdot \phi(x) \geq 0$ as shown in Theorem 2.1. With $(\overline{w}_t, \overline{s}_t)$ following the update in (8) with constant step size $\epsilon$:*

$$\overline{u}_t = u_t(\overline{w}_t, \overline{s}_t)$$
$$\overline{w}_{t+1} = \overline{w}_t - \epsilon\overline{u}_t \circ \overline{v\phi}_t$$
$$\overline{s}_{t+1} = \overline{s}_t + v_t(\overline{w}_t, \overline{s}_t).$$

*Assume $\epsilon > 0$, we have:*

$$\frac{1}{T}\sum_{t=1}^{T}\|\mathcal{L}(\overline{w}_t) \circ \overline{u}_t\|_\phi \leq \frac{\mathcal{L}(\overline{w}_1) - \mathcal{L}(\overline{w}^*)}{T\epsilon} + \frac{\mu\epsilon}{2T}B_T,$$

*where $B_T = \sum_{t=1}^{T}\|\overline{u}_t\|^2$, $\overline{w}^* = argmin_w\mathcal{L}(w)$, and for notions, we write $\|x\|_\phi = x^\top\phi(x), \forall x$.*

*Proof.* Using the $\mu$-smoothness of $\mathcal{L}(\boldsymbol{w})$, we expand $\mathcal{L}(\overline{\boldsymbol{w}}_{t+1}) - \mathcal{L}(\overline{\boldsymbol{w}}_t)$:

$$\mathcal{L}(\overline{\boldsymbol{w}}_{t+1}) - \mathcal{L}(\overline{\boldsymbol{w}}_t) \leq \nabla\mathcal{L}(\overline{\boldsymbol{w}}_t)^\top (\overline{\boldsymbol{w}}_{t+1} - \overline{\boldsymbol{w}}_t) + \frac{\mu}{2} \|\overline{\boldsymbol{w}}_{t+1} - \overline{\boldsymbol{w}}_t\|^2.$$

Substitute $\overline{\boldsymbol{w}}_{t+1} - \overline{\boldsymbol{w}}_t = -\epsilon\overline{\boldsymbol{u}}_t \circ \overline{\boldsymbol{v\phi}}_t$ to get:

$$\mathcal{L}(\overline{\boldsymbol{w}}_{t+1}) - \mathcal{L}(\overline{\boldsymbol{w}}_t) \leq -\epsilon \cdot \nabla\mathcal{L}(\overline{\boldsymbol{w}}_t)^\top \left(\overline{\boldsymbol{u}}_t \circ \overline{\boldsymbol{v\phi}}_t\right) + \frac{\mu\epsilon^2}{2} \left\|\overline{\boldsymbol{u}}_t \circ \overline{\boldsymbol{v\phi}}_t\right\|^2.$$

Simplify using the definition of $\|\cdot\|_\phi$ and $\|\cdot\|_{\Phi_t}$:

$$\nabla\mathcal{L}(\overline{\boldsymbol{w}}_t)^\top \left(\overline{\boldsymbol{u}}_t \circ \overline{\boldsymbol{v\phi}}_t\right) = \|\mathcal{L}(\overline{\boldsymbol{w}}_t) \circ \overline{\boldsymbol{u}}_t\|_\phi,$$

which gives:

$$\mathcal{L}(\overline{\boldsymbol{w}}_{t+1}) - \mathcal{L}(\overline{\boldsymbol{w}}_t) \leq -\epsilon\left(\|\mathcal{L}(\overline{\boldsymbol{w}}_t) \circ \overline{\boldsymbol{u}}_t\|_\phi\right) + \frac{\mu\epsilon^2}{2} \|\overline{\boldsymbol{u}}_t\|^2.$$

Summing over $t = 1, \ldots, T$, we obtain a telescoping sum:

$$\mathcal{L}(\overline{\boldsymbol{w}}_{T+1}) - \mathcal{L}(\overline{\boldsymbol{w}}_1) \leq -\epsilon \sum_{t=1}^{T} \left(\|\mathcal{L}(\overline{\boldsymbol{w}}_t) \circ \overline{\boldsymbol{u}}_t\|_\phi\right) + \frac{\mu\epsilon^2}{2} \sum_{t=1}^{T} \|\overline{\boldsymbol{u}}_t\|^2.$$

Rearranging, dividing by $T\epsilon$, and noting $\mathcal{L}(W_{T+1}) \geq \mathcal{L}(W^*)$, we get:

$$\frac{1}{T} \sum_{t=1}^{T} \|\mathcal{L}(\overline{\boldsymbol{w}}_t) \circ \overline{\boldsymbol{u}}_t\|_\phi \leq \frac{\mathcal{L}(\overline{\boldsymbol{w}}_1) - \mathcal{L}(\overline{\boldsymbol{w}}^*)}{T\epsilon} + \frac{\mu\epsilon}{2T} B_T,$$

where $B_T = \sum_{t=1}^{T} \|\overline{\boldsymbol{u}}_t\|^2$. This concludes the proof. $\qquad\square$

**Theorem A.4.** *Consider updates* (7) *and* (8)*. Assuming $\mathcal{L}(\cdot)$ is $\mu$-smooth and the scaled step size $\epsilon_k\alpha_k \leq \sigma$, and consider the following mask function:*

$$\overline{\boldsymbol{v\phi}}_k = \alpha_k \mathbb{I}(\nabla\mathcal{L}(\overline{\boldsymbol{w}}_k) \circ \overline{vu}_k \geq \frac{\mu\sigma}{2}\overline{vu}_k \circ \overline{vu}_k),$$

*we have*

$$\mathcal{L}(\overline{\boldsymbol{w}}_{k+1}) \leq \mathcal{L}(\overline{\boldsymbol{w}}_k).$$

*which holds for any step size $\epsilon_k \geq 0$.*

*Proof.* By smoothness, we have

$$\mathcal{L}(\overline{\boldsymbol{w}}_{k+1}) - \mathcal{L}(\overline{\boldsymbol{w}}_k)$$

$$\leq \nabla\mathcal{L}(\overline{\boldsymbol{w}}_k)^\top (\overline{\boldsymbol{w}}_{k+1} - \overline{\boldsymbol{w}}_k) + \frac{\mu}{2} \|\overline{\boldsymbol{w}}_{k+1} - \overline{\boldsymbol{w}}_k\|_2^2$$

$$= -\epsilon_k \nabla\mathcal{L}(\overline{\boldsymbol{w}}_k)^\top (\overline{\boldsymbol{v\phi}}_k \circ \overline{\boldsymbol{u}}_k) + \frac{\mu\epsilon_k^2}{2} \|\overline{\boldsymbol{v\phi}}_k \circ \overline{\boldsymbol{u}}_k\|_2^2$$

$$= \epsilon_k\alpha_k \left(-\nabla\mathcal{L}(\overline{\boldsymbol{w}}_k)^\top (\mathbb{I}(\nabla\mathcal{L}(\overline{\boldsymbol{w}}_k) \circ \overline{vu}_k \geq \frac{\mu\sigma}{2}\overline{vu}_k \circ \overline{vu}_k) \circ \overline{\boldsymbol{u}}_k) + \frac{\mu\epsilon_k\alpha_k}{2} \left\|\mathbb{I}(\nabla\mathcal{L}(\overline{\boldsymbol{w}}_k) \circ \overline{vu}_k \geq \frac{\mu\sigma}{2}\overline{vu}_k \circ \overline{vu}_k)\right.$$

$$\leq \epsilon_k\alpha_k \left(-\nabla\mathcal{L}(\overline{\boldsymbol{w}}_k)^\top (\mathbb{I}(\nabla\mathcal{L}(\overline{\boldsymbol{w}}_k) \circ \overline{vu}_k \geq \frac{\mu\sigma}{2}\overline{vu}_k \circ \overline{vu}_k) \circ \overline{\boldsymbol{u}}_k) + \frac{\mu\sigma}{2} \left\|\mathbb{I}(\nabla\mathcal{L}(\overline{\boldsymbol{w}}_k) \circ \overline{vu}_k \geq \frac{\mu\sigma}{2}\overline{vu}_k \circ \overline{vu}_k) \circ \overline{\imath}\right.$$

$$\leq 0.$$

$\qquad\square$

## A.5 INNER PRODUCT MASKS

Out of theoretical interest, let us consider a case of using an inner product mask:

$$\overline{\boldsymbol{v}\phi}_k = \mathbb{I}(\overline{\boldsymbol{u}}_k^\top \overline{\boldsymbol{vg}}_k > 0), \qquad \boldsymbol{vg}_k = \nabla \mathcal{L}(\overline{\boldsymbol{w}}_k),$$

which yields a scalar that applies to the entire update vector. In comparison, the element-wise mask in the paper treats each element separately.

This case is interesting because, on convex functions, the cautious optimizers is always no worse than the base optimizers, *regardless of the step size choices*.

**Theorem A.5.** *Assume $\mathcal{L}(\cdot)$ is convex, and $\overline{\boldsymbol{v}\phi}_k = \mathbb{I}(\overline{\boldsymbol{u}}_k^\top \nabla \mathcal{L}(\overline{\boldsymbol{w}}_k) \geq 0)$. Then, starting from the same point $(\boldsymbol{w}_k, \boldsymbol{s}_k) = (\overline{\boldsymbol{w}}_k, \overline{\boldsymbol{s}}_k)$, we have*

$$\mathcal{L}(\overline{\boldsymbol{w}}_{k+1}) \leq \mathcal{L}(\boldsymbol{w}_{k+1}).$$

*which holds for any step size $\epsilon_k \geq 0$.*

*Proof.* If $\overline{\boldsymbol{u}}_k^\top \nabla \mathcal{L}(\overline{\boldsymbol{w}}_k) > 0$, we have $\overline{\boldsymbol{v}\phi}_k = 1$, and $\boldsymbol{w}_{k+1} = \overline{\boldsymbol{w}}_{k+1}$, and $\mathcal{L}(\overline{\boldsymbol{w}}_{k+1}) = \mathcal{L}(\boldsymbol{w}_{k+1})$.

If $\overline{\boldsymbol{u}}_k^\top \nabla \mathcal{L}(\overline{\boldsymbol{w}}_k) \leq 0$, we have $\overline{\boldsymbol{v}\phi}_k = 0$ and $\overline{\boldsymbol{w}}_{k+1} = \boldsymbol{w}_k$. By the convexity of $\mathcal{L}$, we have

$$\mathcal{L}(\boldsymbol{w}_{k+1}) - \mathcal{L}(\overline{\boldsymbol{w}}_{k+1}) = \mathcal{L}(\boldsymbol{w}_k - \epsilon \boldsymbol{u}_k) - \mathcal{L}(\boldsymbol{w}_k)$$
$$\geq -\epsilon \boldsymbol{u}_k^\top \nabla \mathcal{L}(\boldsymbol{w}_k) > 0.$$

This proves the result. $\square$

**Corollary A.6.** *Consider the elementary test function:*

$$\mathcal{L}(\boldsymbol{w}) = \frac{1}{2} \|\boldsymbol{a} \circ \boldsymbol{w}\|_2^2.$$

*where $\boldsymbol{a} \in \mathbb{R}^d$ is a non-zero vector. This is commonly used as the testbed for optimization algorithms in theoretical analysis.*

*Assume $\boldsymbol{u}_k$ and $\boldsymbol{v}_k$ are element-wise mappings. We have $\mathcal{L}(\overline{\boldsymbol{w}}_{k+1}) \leq \mathcal{L}(\boldsymbol{w}_{k+1})$ given $(\boldsymbol{w}_k, \boldsymbol{s}_k) = (\overline{\boldsymbol{w}}_k, \overline{\boldsymbol{s}}_k)$, with either the inner product mask $\overline{\boldsymbol{v}\phi}_k = \mathbb{I}(\overline{\boldsymbol{u}}_k^\top \nabla \mathcal{L}(\overline{\boldsymbol{w}}_k) \geq 0)$, or the element-wise mask $\overline{\boldsymbol{v}\phi}_k = \mathbb{I}(\overline{\boldsymbol{u}}_k \circ \nabla \mathcal{L}(\overline{\boldsymbol{w}}_k) \geq 0)$.*

*Proof.* The inner product case is implied by Theorem A.5. For element-wise mask, since the loss is an element-wise sum, and the update functions are element-wise mappings, which can apply Theorem A.5 on each element.

This argument, of course, can be extended to general convex and separable loss functions of form $\mathcal{L}(\boldsymbol{w}) = \sum_i \mathcal{L}_i(w_i)$. $\square$

**Theorem A.7.** *Consider update (8). Assuming $\mathcal{L}(\cdot)$ is $\mu$-smooth, and consider the following mask function:*

$$\overline{\boldsymbol{v}\phi}_k = \alpha_k \mathbb{I}(\nabla \mathcal{L}(\overline{\boldsymbol{w}}_k)^\top \overline{\boldsymbol{vu}}_k \geq \frac{\alpha_k \mu \epsilon_k}{2} \|\overline{\boldsymbol{vu}}_k\|^2),$$

*where $\alpha_k$ is a scaling factor.*

*We have*

$$\mathcal{L}(\overline{\boldsymbol{w}}_{k+1}) \leq \mathcal{L}(\overline{\boldsymbol{w}}_k).$$

*which holds for any step size $\epsilon_k \geq 0$.*

**Ablation on $\phi$:** To analyze the impact of cautious masking, we conduct ablation studies on different choices of $\phi$. Our results show that the performance is not sensitive to the specific choice of $\phi$, as all cautious variants incorporating these functions consistently outperform GDM. This underscores the robustness and effectiveness of cautious masking, as shown in Figure 3.

The tested $\phi$ functions are defined as follows:

$$\phi_c = \mathbb{I}(x > 0) - c\mathbb{I}(x \leq 0),$$
$$\phi_{\texttt{inner}} = \mathbb{I}(x^\top y \geq 0). \tag{13}$$

These results highlight that cautious masking enhances optimization stability and performance across diverse settings.

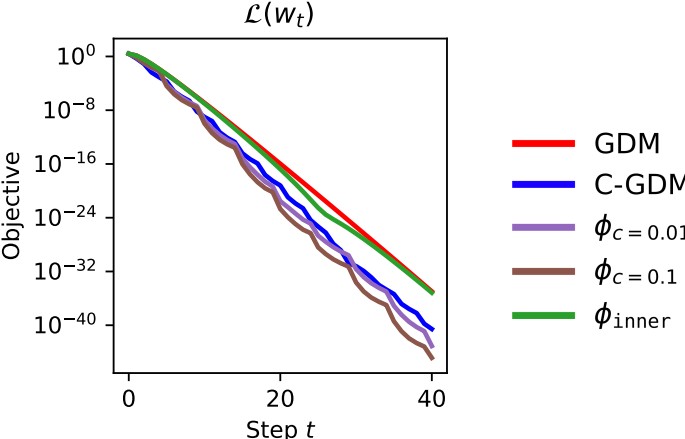

Figure 3: This plot presents an ablation study evaluating the performance of various $\phi$ configurations. We compare the loss curves for different $\phi$ choices. Across all tested configurations, the cautious GDM variants, $\phi_c$ with $c = 0.01, 0.1$, and $\phi_{\texttt{inner}}$ as defined in (13), consistently outperform standard GDM with optimal hyper-parameter configration (Goh, 2017).

## A.6 Pesudo Code

See Algorithm 3 for the Cautious variant of the Lion optimizer.

---

**Algorithm 3** C-Lion Optimizer

---

**Require:** learning rate $\epsilon$, momentum coefficient $\beta_1, \beta_2 \in [0, 1)$, weight decay factor $\gamma$
1: Initialize parameter vector $\boldsymbol{w}_t$
2: Initialize $t = 0$, $\boldsymbol{m}_0 = \boldsymbol{0}$
3: **while** $\boldsymbol{w}_t$ not converged **do**
4:    $t \leftarrow t + 1$
5:    $g_t \leftarrow \nabla_{\boldsymbol{w}} \mathcal{L}_t(\boldsymbol{w}_{t-1})$                                    {Get gradients at timestep $t$}
6:    $u_t \leftarrow \mathrm{sign}(\beta_1 m_{t-1} + (1 - \beta_1) \cdot g_t)$                {get the signed update}
7:    $m_t \leftarrow \beta_2 m_{t-1} + (1 - \beta_2) \cdot g_t$                      {update momentum}
8:    $\boldsymbol{\phi}_t \leftarrow \mathbb{I}(\boldsymbol{u}_t \circ \boldsymbol{g}_t \geq 0)$                         // Compute alignment mask
9:    $\bar{\epsilon}_t = \epsilon_t \frac{d}{\|\boldsymbol{\phi}_t\|_0 + 1}$                         // Scale lr, $d$ is dimension of $\boldsymbol{\phi}_t$
10:   $\boldsymbol{w}_t \leftarrow \boldsymbol{w}_{t-1} - \bar{\epsilon}_t \boldsymbol{\phi}_t \circ \boldsymbol{u}_t$    // Masked update
11:   $\boldsymbol{w}_t \leftarrow \boldsymbol{w}_t - \epsilon \gamma \boldsymbol{w}_t$                            {Weight decay}
12: **end while**

---

# B Cautious Mask Study

Figure 4 shows the ratio of active dimensions $r_k = \mathrm{nnz}(\overline{\boldsymbol{vu}}_k \circ \overline{\boldsymbol{vg}}_k > 0)/\dim(\overline{\boldsymbol{vu}}_k \circ \overline{\boldsymbol{vg}}_k)$ across the training iteration $k$ on LLaMA 100M. It see that it decreases from 1 and stabilizes around 0.55. This suggests that the scaling factor generally lies within the range $[1, 1.55]$. The stable ratio does not change significantly across different models.

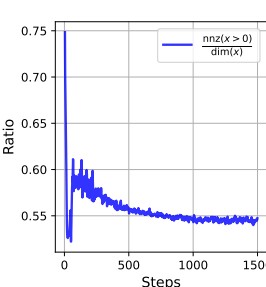

Figure 4: The sparsity ratio $r(\mathbf{x}) = \frac{\mathrm{nnz}(\mathbf{x} > 0)}{\dim(\mathbf{x})}$ during pretraining of LLaMA 100M on the C4 dataset using the C-AdamW optimizer. The ratio quantifies the proportion of nonzero elements in the representations over training steps.

# C Additional Details on LLM Scaling

The training sequence length is set to be 4096 and we perform coordinate-descent over discrete grids for all optimizer hyperparameters (lr, weight decay, warmup, $\beta_1$, $\beta_2$, $\epsilon$, max-grad-norm, batch size). The following tables 5, 6 show the optimal set of parameters for each scale.

# D Additional Details on Mini-ImageNet

Here you can find the hyperparameter search of the Mini-ImageNet experiments. The findings are consistent with our LLM experiments.

| Hyperparameter | AdamW | | | | C-AdamW | | | |
|---|---|---|---|---|---|---|---|---|
| | 130M | 300M | 520M | 1.2B | 130M | 300M | 520M | 1.2B |
| Learning rate (lr) | 0.008 | 0.008 | 0.004 | 0.002 | 0.008 | 0.008 | 0.008 | 0.006 |
| Weight decay (wd) | 0.1 | 0.1 | 0.2 | 0.2 | 0.1 | 0.1 | 0.1 | 0.1 |
| Warmup steps | 2000 | 2000 | 1000 | 1000 | 2000 | 2000 | 2000 | 2000 |
| $\beta_1$ | 0.9 | 0.9 | 0.9 | 0.9 | 0.95 | 0.98 | 0.98 | 0.98 |
| $\beta_2$ | 0.98 | 0.98 | 0.98 | 0.98 | 0.98 | 0.98 | 0.98 | 0.98 |
| $\epsilon$ | 1e-20 | 1e-10 | 1e-10 | 1e-05 | 1e-15 | 1e-25 | 1e-25 | 1e-16 |
| Max-grad-norm | 1 | 1 | 1 | 2 | 1 | 2 | 1 | 1 |
| Batch size | 128 | 128 | 256 | 256 | 128 | 128 | 256 | 256 |

Table 5: Optimal hyperparameters (from coordinate-descent over discrete grids) for AdamW and C-AdamW across model scales.

| Hyperparameter | Range |
|---|---|
| Learning rate (lr) | {0.001, 0.002, 0.004, 0.006, 0.008, 0.010, 0.012, 0.014, 0.016, 0.018, 0.020} |
| Weight decay (wd) | {0.05, 0.1, 0.15, 0.2, 0.25, 0.3, 0.35, 0.4} |
| Warmup steps | {500, 1000, 2000, 4000, 6000} |
| $\beta_1$ | {0.85, 0.90, 0.92, 0.95, 0.98, 0.99} |
| $\beta_2$ | {0.96, 0.98, 0.99, 0.995, 0.9995} |
| $\epsilon$ | {1e-30, 1e-25, 1e-20, 1e-15, 1e-10, 1e-5} |
| Max-grad-norm | {0.5, 1, 2, 4} |
| Batch size | {32, 64, 128, 256, 512} |

Table 6: Discrete grid for hyperparameter search

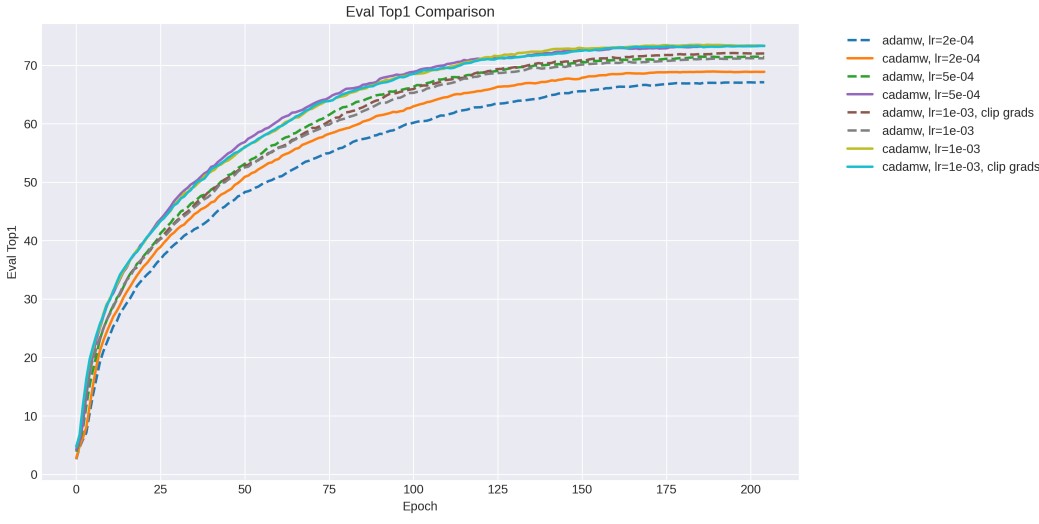

Figure 5: AdamW/C-AdamW learning rate search on Mini-ImageNet

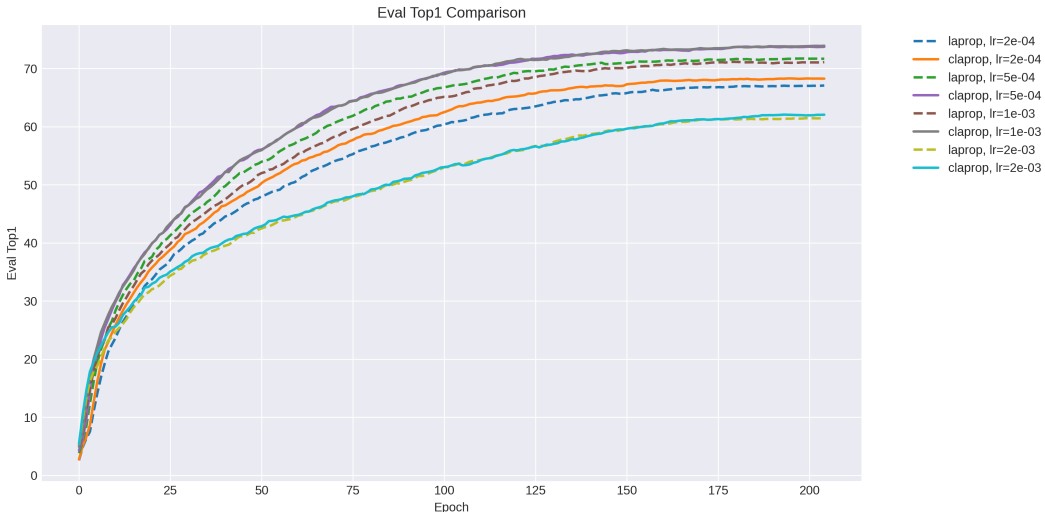

Figure 6: LaProp/C-LaProp learning rate search on Mini-ImageNet

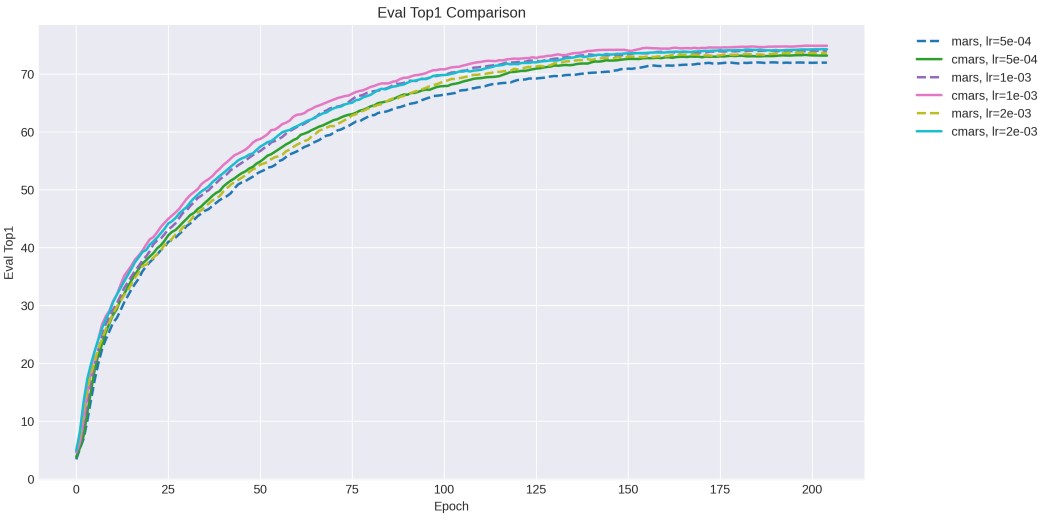

Figure 7: MARS/C-MARS learning rate search on Mini-ImageNet

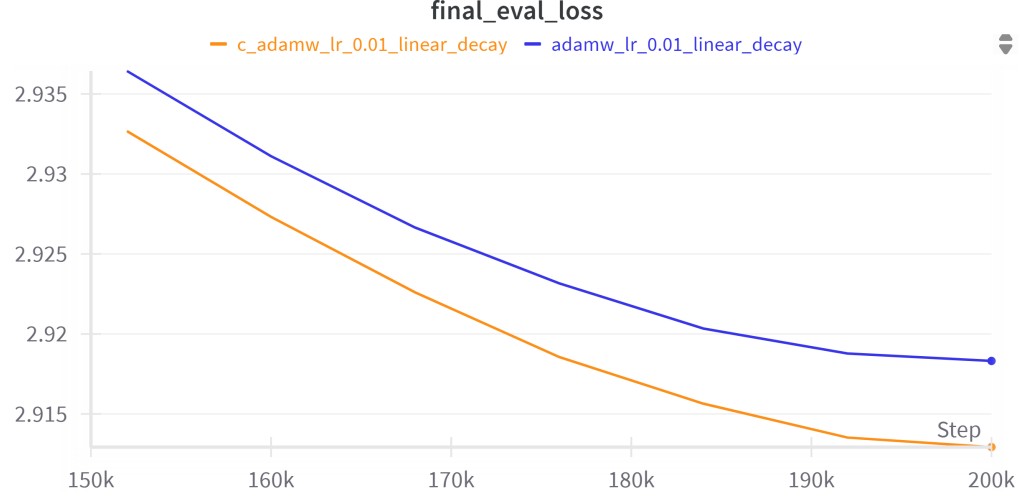

Figure 8: This experiment follows the same setup as the previous 100M Llama experiment in 1

# E    ADDITIONAL EXPERIMENTS

## E.1    LEARNING RATE SCHEDULER ABLATION

We notice a concurrent work (Bergsma et al., 2025) suggests that reducing learning rate straight to zero with linear decay is beneficial. To preliminarily validate the robustness of cautious optimizer, we took the 100M model setup in our LLM pretraining experiment and swap in the new scheduler. We found cautious optimizers stably outperform baselines.

## E.2    BATCH SIZE ABLATION ON 60M

We take the 60M parameter model and perform a sweep on Batch Sizes and train for 1.2 billion tokens ($1\times$ Chinchilla) on C4 and we are reporting perplexity (lower the better).

| Batch Size (tokens) | 24K | 120K | 600K |
|---|---|---|---|
| AdamW | 38.5 | 37.2 | 41.7 |
| C-AdamW | 37.2 | 36.2 | 40.6 |

Table 7: Perplexity of AdamW vs. C-AdamW across different batch sizes.

## E.3    MUON (JORDAN ET AL., 2024)

We tested a preliminary version of C-Muon (Fig 9) and found that results encouraging as follows. Although at the time of submission we are not able to verify it at larger scale, it could potentially be a very interesting future direction.

## E.4    POST-TRAINING LLM

To further test cautious optimizers in language modeling tasks, we perform two post-training experiments. First, we instruction-tune Qwen2.5-1.5B-Instruct (Yang et al., 2024) on the PowerInfer/QWQ-LONGCOT-500K (PowerInfer, 2024), a dataset focusing on enhancing the model's reasoning abilities by data distilled from QwQ-32B-Preview (Qwen, 2024). Then we follow the experiment setting in (Huang et al., 2024) to perform RLHF alignment with PPO (Schulman et al., 2017) on EleutherAI/pythia-1b-deduped.

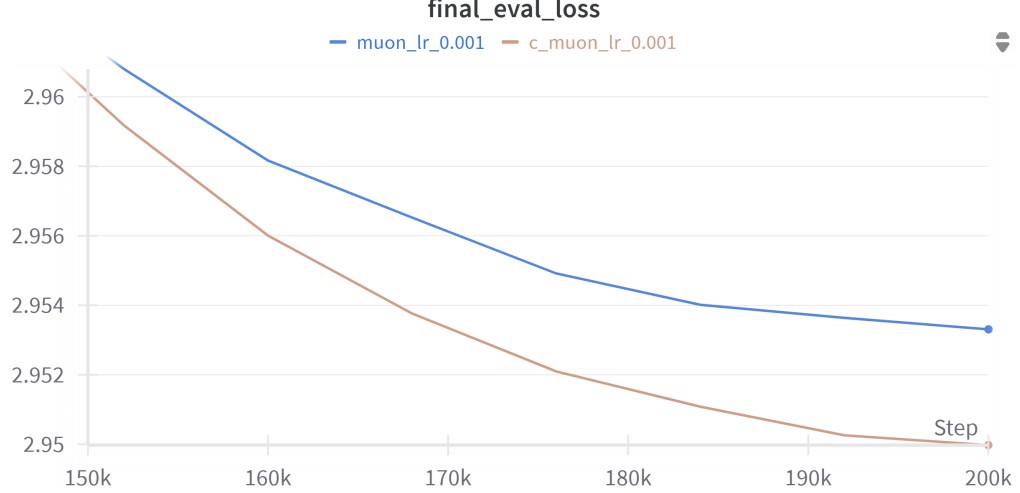

Figure 9: Following the same 100M model training setup in 1, we also preliminarily test the latest popular optimizer Muon Jordan et al. (2024) and find cautious improves upon the baseline.

**Observation:** In both experiments, we find under same training steps and PPO episodes, cautious optimizers obtain lower training loss as well as higher rewards. This indicates potential of our proposed method beyond pretraining tasks.

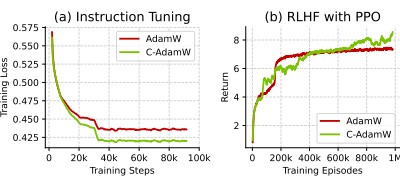

Figure 10: (a) shows training loss on the task of instruction finetuing Qwen2.5-1.5B-instruct on the distilled dataset from QwQ for 500K question and answer pairs. (b) shows RLHF reward with PPO for 1 million episodes. In both cases, C-AdamW outperforms AdamW.

### E.5 PRETRAINING MASKED AUTOENCODERS (MAEs)

Masked Autoencoders (MAEs) (He et al., 2022) have emerged as a powerful approach for pretraining Vision Transformers (ViTs) (Dosovitskiy, 2020) on large-scale datasets like ImageNet-1K (Russakovsky et al., 2015). This task involves reconstructing 75% of randomly masked image patches, a challenging objective that requires extensive training over hundreds of epochs and millions of images. The primary goal is to learn robust visual representations that are generalizable across downstream vision tasks. The quality of these representations is typically measured by the final evaluation loss, which reflects how accurately the model reconstructs masked test images; lower evaluation loss indicates higher-quality representations. The results of our experiments are summarized in Figure 11, where we compare the performance of the cautious optimizer against the AdamW baseline.

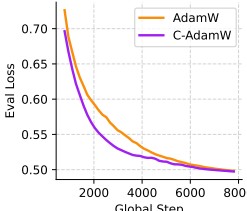

Figure 11: Evaluation loss of pretrained MAEs on ImageNet1K on ViT backbone for 50 epochs, using AdamW and C-AdamW. Hyperparameters can be found in Table 8.

**Observation:** From figure 11, we observe that the cautious optimizer achieves lower evaluation loss faster compared to AdamW. This result highlights the effectiveness of the cautious approach

| # Params | $\beta_1$ | $\beta_2$ | Learning rate | weight decay | Batch Size |
|----------|-----------|-----------|---------------|--------------|------------|
| 110 M | 0.9 | 0.999 | $1.5\times10^{-4}$ | 0.05 | 4096 |

Table 8: Hyperparameters for MAE experiment. This experiment follows the optimal setup provided in He et al. (2022)

in improving the precision of reconstruction and, consequently, the quality of the learned visual representations.

## F  MISCELLANEOUS

The masking and scaling would incur some additional costs. However, in practice we observe that the impact is minimum to overall throughput in the Distributed Data Parallel setting. For our 100M model runs on 16 GPUs, AdamW has token throughput of 579383 token/s, whereas C-AdamW has 551483 tokens/s, which is around 3% difference in training throughput. Note that this is comparing our naive cautious implementation against the fused pytorch implementation.

As for other distributed training setting, such as tensor parallel, masking operation is element-wise hence not communication-bound, whereas scaling would require global statistics. However, the communication is also minimum, since only a single floating-point number (the local mean) needs to be all-gathered by other workers. Given modern GPU bandwidth, this should incur only minor overhead.

Table 9: Training efficiency comparison between AdamW and C-AdamW.

| Method | Wall-Clock (h) | Throughput (token/s) |
|--------|----------------|----------------------|
| AdamW | 10.547 | 571,839.76 |
| C-AdamW | 10.567 | 560,488.63 |

