# OpenReview forum: "Cautious Optimizers: Improving Training with One Line of Code"
_ICLR.cc/2026/Conference — ICLR 2026 Poster_

### Official Review · Reviewer_uTNE · 2025-10-15

**Soundness:** 4
**Presentation:** 4
**Contribution:** 4
**Rating:** 8
**Confidence:** 4

**Summary:**

The paper presents "Cautious Optimizers," a simple, one-line modification for momentum-based optimizers like AdamW. The method only applies parameter updates when their direction aligns with the sign of the current gradient, preventing counter-productive steps. This change is claimed to accelerate model training (including for LLMs and computer vision) with minimum extra tuning on hyperparameters, all while preserving theoretical convergence guarantees under the Lyapunov analysis.

**Strengths:**

- The change is trivial (a single line of code) and applicable to existing momentum-based optimizers.

- The method boosts performance without the need for costly and time-consuming hyperparameter retuning.

- The intuitive idea is supported by theoretical analysis, ensuring that convergence guarantees are maintained.

- It demonstrates consistent speed-ups in various high-impact domains, including LLMs and image classification.

**Weaknesses:**

N/A

**Questions:**

N/A

---

> ### Author Response · Authors · 2025-11-16
> **Thank you for your support!**
>
> Thank you for the thoughtful and encouraging review. We greatly appreciate your positive assessment of the paper’s soundness, clarity, and contribution.

---

> > ### Public Comment · ~Nan_Qiao1 · 2026-04-14
> >
> > Great paper!

---

### Official Review · Reviewer_G1tn · 2025-10-18

**Soundness:** 3
**Presentation:** 3
**Contribution:** 3
**Rating:** 6
**Confidence:** 4

**Summary:**

The paper proposes a simple modification to momentum-based optimizers called Cautious Optimizers, implemented by masking update directions that disagree in sign with the current gradient. The authors argue that this one-line change ensures monotonic decrease in loss for small step sizes, preserves the Lyapunov/Hamiltonian structure of momentum dynamics, and empirically improves convergence speed and training stability. Experiments on toy problems, LLM pretraining (100M LLaMA on C4 and FineWeb-Edu), and vision tasks (Mini-ImageNet with ViT) confirm that the proposed modification can improve the performance of some popular optimizers.

**Strengths:**

- The paper addresses a practically relevant problem.

- The proposed method is very simple (a one-line change in PyTorch) and broadly applicable to many optimizers.

- The experiments show consistent (but modest) gains in LLM pretraining, image classification, and toy settings, with improved stability and slightly better downstream performance.

- The theoretical analysis tries to conceptually connect the masking heuristic to theoretical convergence guaranties.

**Weaknesses:**

1. Theoretical claims are not fully convincing. The results demonstrate that cautious optimizers can reduce the loss more than the original optimizers in a *single step*, but not throughout the full optimization trajectory. In line 84, the authors claim that "Our theoretical analysis shows that the modified algorithm converges to local optima under mild conditions on the base optimizers", but the presented results do not establish such convergence.

2. Lack of stochastic analysis (the paper only considers deterministic gradient setting).

3. Empirical improvements are modest. Most improvements (e.g., Table 2) are around $0.1–1$%, which may fall within variance across training runs.

4. The abstract claims "consistent speed-up", but training time or throughput overhead is not reported (the results focus on perplexity and accuracy).

5. No results are shown on other architectures (CNNs, RNNs) or non-transformer tasks.

6. Some equations in the appendix overflow the page margins.

7. Minor issues: several typos and grammatical errors (e.g., 'methods normally requires' in line 43, 'The follow is a comparison result' in line 240, 'catuious' in line 390, 'generalit' in line 444),  inconsistent boldface notation (e.g., equation (1)), formatting issues in References.

**Questions:**

1. Could the authors address the concerns above?

2. In Table 1, why were C-AdamW runs with learning rates 1e-4 and 3e-4 omitted?

---

> ### Author Response · Authors · 2025-11-16
> **Responses to Reviewer G1tn's Questions (Part I)**
>
> We appreciate Reviewer G1tn’s attentiveness to details. We will address each concern point by point below.
> ## Theoretical claims
> Our core theoretical claim is conservative: we merely show that cautious masking is a legitimate operation and it does not lead to divergence, given that the base optimizer itself converges in the first place. The tool we use is classic Lyapunov analysis, which shows the upper bound of the loss function is a strictly decreasing function wrt optimization steps/time. So our results DO establish convergence for both the base optimizers (AdamW, Lion and etc) as well as their cautious variants. Our conditions/assumptions are also conventional in literature, requiring only smoothness and differentiability.
>
> What we DID NOT show and left for future works was establishing “faster convergence rate” over multiple steps. Frankly speaking, we found that it was very difficult to show theoretically “strictly faster” convergence over the base optimizers beyond a single step. Intuitively, taking a locally optimal step does not always guarantee that it moves towards the global optimum, especially given the complex loss landscape of modern deep learning problems. Instead, we had to demonstrate it empirically.
>
> ## Re: Empirical improvements
> Pretraining experiments in general have very small variance, especially in the compute optimal regime with exhaustively tuned hyperparameters. We would argue that the improvement is solid and consistent at least in the scale we could afford. Some recent paper [1] also shows a similar range of improvement on many of the newly proposed optimizers once the baseline is properly tuned.
>
> [1] Wen, Kaiyue, David Hall, Tengyu Ma, and Percy Liang. "Fantastic pretraining optimizers and where to find them." arXiv preprint arXiv:2509.02046 (2025).
> ## Re: Speedup
> The speedup in LLM training in our claims referred to tokens seen reaching the same perplexity. To get a wall clock speedup under fair comparison (e.g comparing against fused AdamW in pytorch), it involves system level implementation details such as fused/parallel applied versions of cautious optimizers, which is a bit beyond the scoop of a machine learning paper. It also over complicates the comparisons since different distributed training strategies have different implications (FSDP or Tensor Parallel).  Nonetheless, in the appendix section F, we provide e2e training throughput comparison between our naive cautious implementation against fusedAdamW.
>
>
> ## Re: Other architectures
> We chose to focus on transformers, since it’s the dominant architecture in the field right now. Nonetheless, we believe it should also generalize to other architectures. Due to time and resource constraints, we can only show some small additional experiments here, bigger and more rigorous experiments would have to be left for future studies.
>
> On Cifar10, a LeNet-5 (3 conv layers and 2 fully connected layers) trained from scratch
> | Method  | train loss        | validation acc |
> |-------|-------------|--------------------|
> | AdamW | 1.0050  | 69.78%              |
> | C-AdamW  | 0.8977   | 72.38%     |
>
> On Shakespear character-level LSTM
> | Method  | train loss        | validation ppl |
> |-------|-------------|--------------------|
> | AdamW | 1.6723  | 5.83              |
> | C-AdamW  | 1.6514   | 5.68     |
>
> ## Re: Why omitting 1e-4 and 3e-4?
> On the baseline AdamW, these two learning rates are too small and clearly outside of the optimal regime.
>
> ## Re: Format and Typos
> Thanks for pointing those out, we will make sure to fix them in the final version.

---

> > ### Author Response · Authors · 2025-11-16
> > **Responses to Reviewer G1tn's Questions (Part II)**
> >
> > ## Re: Stochastic Analysis
> > Stochastic analysis on AdamW itself is a hard topic without strong assumptions. Here we provide an analysis for cautious SGD with momentum, which is a straightforward extension over standard SGD with momentum analysis.
> >
> > Considering following update rules:
> > $$v_{t+1} = \beta v_t + g_t,$$
> > $$A_{t,i} = \mathbf{1} \\{ g_{t,i} \cdot v_{t+1,i} \ge 0 \\}, \qquad A_t \in {0,1}^d,$$
> > $$\theta_{t+1} = \theta_t - \eta_t , (A_t \odot v_{t+1}),$$
> >
> > where $ A_t \odot v_{t+1} $ is the elementwise product.
> >
> > ---
> >
> > ### Assumptions
> >
> > We assume:
> >
> > 1. **$L$-smoothness.**
> >    $$
> >    f(y) \le f(x) + \langle \nabla f(x), y-x\rangle + \frac{L}{2}|y-x|^2.
> >    $$
> >
> > 2. **Lower boundedness.**
> >    $$ f_\star := \inf_\theta f(\theta) > -\infty. $$
> >
> > 3. **Unbiased gradients with bounded variance.**
> >    $$
> >    \mathbb{E}[g_t \mid \mathcal{F}_t] = \nabla f(\theta_t), \qquad
> >    \mathbb{E}[|g_t|^2 \mid \mathcal{F}_t] \le G^2.
> >    $$
> >
> > 4. **Stepsizes.**
> >    $$
> >    \eta_t > 0,\quad \sum_t \eta_t = \infty,\quad \sum_t \eta_t^2 < \infty.
> >    $$
> >
> > 5. **Bounded momentum.**
> >    $$
> >    \mathbb{E}[|v_{t+1}|^2 \mid \mathcal{F}_t] \le C_1.
> >    $$
> >
> > 6. **Coordinate-wise non-degenerate cautiousness.**
> >    There exist $ \kappa>0 $, $ C_0\ge 0 $ such that
> >    $$
> >    \mathbb{E}\left[
> >    \sum_{i=1}^d A_{t,i} \nabla_i f(\theta_t) v_{t+1,i}
> >    \middle| \mathcal{F}_t
> >    \right]
> >    \ge
> >    \kappa |\nabla f(\theta_t)|^2 - C_0.
> >    $$
> >    This assumption guarantees that cautious mask is reasonable and not always zero. It is the fundamental new assumption needed in cautious analysis and does not appear in standard analyses.
> > ---
> > ### Descent Inequality
> >
> > Using $ L $-smoothness with
> > $$
> > \Delta_t = -\eta_t (A_t \odot v_{t+1}),
> > $$
> > we have
> > $$
> > f(\theta_{t+1})
> > \le
> > f(\theta_t)
> >  \langle \nabla f(\theta_t), \Delta_t\rangle
> >   +\frac{L}{2}|\Delta_t|^2.
> >   $$
> >
> > That is,
> > $$
> > f(\theta_{t+1})
> > \le
> > f(\theta_t)
> > -\eta_t \sum_{i=1}^d A_{t,i} \nabla_i f(\theta_t) v_{t+1,i} + \frac{L}{2}\eta_t^2 |A_t \odot v_{t+1}|^2. $$
> >
> > Taking conditional expectation and applying bounded-variance & bounded-second-moment assumptions,
> >
> > $$
> > \mathbb{E}\left[
> > f(\theta_{t+1}) \mid
> > \mathcal{F}_t
> > \right]
> > \le
> > f(\theta_t) + \kappa \eta_t \|\nabla f(\theta_t)\|^2 - C \eta_t
> > $$
> >
> > where $ C $ absorbs the $ \eta_t^2|v_{t+1}|^2 $ term.
> >
> > ---
> >
> > ### Almost–Supermartingale Structure
> >
> > Define
> > $$
> > X_t := f(\theta_t)-f_\star + \sum_{k\ge t} C\eta_k.
> > $$
> >
> > Then $ X_t\ge 0 $ and
> > $$
> > \mathbb{E}[X_{t+1} \mid \mathcal{F}_t]
> > \le
> > X_t - \kappa \eta_t |\nabla f(\theta_t)|^2.
> > $$
> >
> > By the **Robbins–Siegmund theorem**, it follows that:
> >
> > $ X_t $ converges almost surely, and
> >   $$
> >   \sum_{t=0}^\infty \eta_t |\nabla f(\theta_t)|^2 < \infty
> >   \quad\text{a.s.}
> >   $$
> >
> > ---
> >
> > ### Convergence
> >
> > Since $ \sum_t \eta_t = \infty $,
> > $$
> > \liminf_{t\to\infty} |\nabla f(\theta_t)| = 0
> > \quad\text{a.s.}
> > $$
> >
> > Thus **every accumulation point** of $ \theta_t $ is a stationary point of $ f $.
> > Cautious SGD with momentum therefore retains the convergence guarantees, provided the coordinate-wise non-degeneracy condition holds.
> >
> > ---

---

### Official Review · Reviewer_4Ez6 · 2025-10-30

**Soundness:** 2
**Presentation:** 2
**Contribution:** 2
**Rating:** 4
**Confidence:** 4

**Summary:**

The paper proposes “Cautious Optimizers,” a one-line masking of momentum updates that zeros coordinates where the proposed update and current gradient have opposite signs, optionally rescaled by the active-mask ratio (Alg. 1). The theory is framed via a Hamiltonian+Descent view (continuous time) and per-step comparison results (discrete time). Experiments include a 2-D toy, Mini-ImageNet, and LLM pretraining up to 1.2B parameters.

**Strengths:**

1. This paper proposes a simple mechanism (coordinate-wise sign check) that is easy to implement; clear statement that it promotes monotone decrease of the loss for small steps.


2. Empirical results cover both vision and language, and generally show small but consistent improvements

**Weaknesses:**

1. The key idea of masking coordinates where the gradient and velocity have opposite signs is meant to promote descent. But requiring sign consistency on every coordinate feels too strict — usually, it's enough for the update direction and gradient to have a positive inner product. Also, in stochastic settings with noisy gradients, enforcing per-coordinate alignment could hurt rather than help. Plus, the paper motivates this with “monotonic decrease,” but per-step monotonicity isn’t necessary for faster convergence — e.g., Nesterov's method is non-monotone. This raises the question: is this motivation really essential?

2. In Theorem 2.3 and 2.4, the discrete-time analysis relies on certain properties of the masking function (like Δ(vₓ) ≥ 0). But Algorithm 1 uses a non-smooth hard indicator (1(uᵢgᵢ > 0)) and a heuristic rescaling. These look inconsistent. Is this difference purely due to the hard masking being non-smooth? If so, it would be good to clarify the gap between what’s proved and what’s implemented.

3. The toy experiment in §3.1 is just 2D and too simple. The LLM results in §3.2 only go up to 1.2B parameters, which is relatively small. To claim relevance to large-scale pretraining, results on models at 7B scale or above would be much more convincing.


4. It feels odd to postpone related work to the very end (§4). This makes §2.1 hard to follow since many readers won’t be familiar with the Hamiltonian perspective. I'd recommend moving related work earlier or giving at least a brief summary in §2.

**Questions:**

See comments 2 of weaknesses.

---

> ### Author Response · Authors · 2025-11-16
> **Responses on Motivation, Theory, and Experimental Setup**
>
> We are happy to answer reviewer 4Ez6’s insightful questions.
> ## Re: Motivation
> “requiring sign consistency on every coordinate feels too strict”, this is true if the optimizer had discarded the update direction completely. However in the case of Cautious, it simply delays the updates (by accumulating it into the momentum). If it’s truly a persistent direction, it would be reflected in the later updates. This turns out to be rather beneficial in the stochastic noisy setting such as reinforcement learning (some preliminary experiments can be found in Appendix E.4). We speculate that it serves as “variance reduction” implicitly and it will be interesting to study this in future works when we have enough resources.
>
> We agree that “per-step monotonicity isn’t necessary for faster convergence”. This turns out to be extremely difficult to show theoretically, and we don’t know if that is ever possible without stronger assumptions. That is why we compromised to provide only proof for single-step and then validate multi-step empirically.
>
>
> ## Re: Δ(vₓ) ≥ 0 and Δ(vₓ) > 0
> Theorem 2.3 and 2.4 are both assuming Δ(vₓ) > 0 (eq 7, eq 8). The smoothness assumption is not on the indicator but the loss function. Please kindly refer to the proof in Appendix section A4, the ≥ is simply an artifact when back-calculating step size. It can be swapped with > if strict decrease on the loss function is desired.
>
> We also exclusively use Δ(vₓ) > 0 in practice, though empirically we’ve found no difference between the two implementations.
>
>
> ## Re: Scale of experiments
> The results in table 2 are obtained via extensive hyperparameter search (learning rate, weight decay, warmup steps, beta1, beta2, esp, max grad norm as well as batch size) across 4 different scales trained up to 20x bpp (Chinichilla Optimal). Please refer to Appendix C table 5 and table 6 for details.
>
> We’ve attempted some 7B runs, but given the model size, it’s difficult to show results with confidence at the same level of rigor (extensive parameter search as above + 140B token budgets), since we do not have that many GPU resources. Nonetheless, we see similar phenomena in the few runs with hand picked hyperparameters. Since these hyperparameters could be suboptimal, we simply do not feel comfortable including the results in the paper.
>
>
> ## Re: Related Works
> Thanks for the suggestion. We will take a closer look to adjust the order accordingly.

---

> > ### Comment · Reviewer_4Ez6 · 2025-11-28
> >
> > Thank you for the responses. I also tried your algorithm in some training tasks, it behaves better than Adam.
> > So, I will change my evaluation to be positive.

---

### Official Review · Reviewer_CSiD · 2025-11-01

**Soundness:** 2
**Presentation:** 3
**Contribution:** 2
**Rating:** 6
**Confidence:** 3

**Summary:**

The authors introduce cautious optimizers which is a modification applicable to any momentum-based optimizer where the main idea is to update parameters mainly when the proposed update direction and the gradient have the same sign. The authors show theoretically that it preserves the optimizer’s Hamiltonian structure and guarantees monotonic decrease of the loss under sufficiently small steps. Empirically, C-AdamW, C-Lion, etc. yield performance improvements in large-scale pretraining and image classification.

**Strengths:**

- The idea is quite elegant can be seamlessly applied to all momentum-based optimizers without introducing new hyperparameters.

- The authors situate the method within the Hamiltonian Descent framework, which were used to analyze Adam and Lion. Theorem 2.1 and Corollary 2.2 offers a clear interpretation of cautious masking as energy-preserving damping.

- The discrete-time results confirm that each cautious step decreases the loss more efficiently than the base optimizer under µ-smoothness.

- The experiments show improvements without re-tuning hyper-parameters — showing that the cautious modification preserves hyperparameter stability.

- Overall, writing in the paper is clear.

**Weaknesses:**

- The claim that cautious optimizers “do not get stuck at non-stationary points even when the update is fully masked out” is only partially justified. The authors state that momentum dynamics will eventually realign updates and gradients, but did not provide no empirical analysis of how long this alignment requires. Maybe exploring the failure cases near saddle points or flat regions (regimes that dominate the modern deep learning optimization) could help?

- Empirically, the improvements are modest and sometimes fall within the noise range of large-scale training. For example, Table 2 reports ≤ 1 % perplexity gains at max. The comparison is also relatively limited i.e., Section 4 mainly discusses related optimizers like AdamW, Lion, but omits several recent and directly relevant baselines such as AdaBelief, Adan and SOAP even when these methods employ simple directional or normalization modifications.

- the effect of the scaling factor introduced in Eq. (1) on the magnitude of updates and on the effective learning-rate distribution is not thoroughly analyzed. Since α directly scales updates in proportion to the ratio, could it change the convergence behavior in anisotropic curvature regions?

**Questions:**

Please refer to the comments in the weaknesses section.

---

> ### Author Response · Authors · 2025-11-16
> **Responses to momentum gradient alignment, improvement range and effect of scaling**
>
> We are humbled that reviewer CSiD found our work “elegant and clear”.
> ## Re: Alignment between masks and gradients
> Our claim does not rely on empirical heuristics: even when the cautious mask zeroes the step, the momentum buffer continues to update with the current stochastic gradient. Unless the gradient remains exactly orthogonal to the buffer indefinitely (a measure-zero event), the buffer realigns and the mask lifts, so the optimizer cannot stay stuck at a non-stationary point.
>
> Saddle points and flat regions do not change this behavior: once the mask lifts, the dynamics reduce to standard momentum, which is known to escape saddles efficiently. While analyzing exact escape times is beyond our current scope, the mask can only delay, not prevent, alignment, and stochastic gradients in deep networks make persistent degeneracy extremely unlikely.
>
> To further prove the above, we have also included a small study on the mask evolution in Appendix Section B figure 4. It’s shown that after warmup, the mask alignment stabilizes around 0.55.
> ## Re: Empirical improvement
> Pretraining runs generally show very low variance, especially in the compute-optimal setting where hyperparameters are carefully tuned. In this regime, even small gains tend to be reliable, and we consistently observe our improvements across repeated runs within our compute budget. While larger-scale experiments would further reduce uncertainty, the trend is stable in all cases we tested.
>
> This observation is also consistent with recent work. Wen et al. [1] report that, once the baseline optimizer is properly tuned, many newly proposed optimizers show improvements in a similar range. This supports our claim that the gains we report are real and not the result of noise or under-tuned baselines.
>
> [1] Wen, Kaiyue, David Hall, Tengyu Ma, and Percy Liang. Fantastic Pretraining Optimizers and Where to Find Them. arXiv:2509.02046 (2025).
> ## Re: Latest Optimizers
> Thank you for highlighting these optimizers. We will expand our discussion of them in the related-work section. Preconditioned methods such as SOAP are largely orthogonal to our approach, and in principle can be composed with it. Separately, we have conducted preliminary experiments with Muon—a special case of Shampoo, which itself generalizes SOAP—and include the results in Appendix E.3 (Figure 9). Notably, the cautious variant of Muon shows early but consistent improvements over the standard Muon baseline.
>
>
> ## Re: Scaling factor
> Theoretically, the Hamiltonian stays the same after applying the scaling factor. Hence, the convergence guarantee is maintained.
>
>
> Empirically, the scaling factor serves as an attempt to maintain the overall update norm.  Appendix section B figure 4 shows thatscaling factor stablizes quickly after warmup.

---

### Official Review · Reviewer_TurQ · 2025-11-03

**Soundness:** 3
**Presentation:** 3
**Contribution:** 3
**Rating:** 6
**Confidence:** 3

**Summary:**

This paper proposes a plug-in method for improving momentum-based optimization methods (such as AdamW) named as Cautious optimizers. This method applies a coordinate-wise mask according to if the proposed update direction aligns in sign with the current gradient. Theoretically, the authors analyze the method in a proposed Hamiltonian/Lyapunov framework. With smoothness-based arguments, they show that cautious optimizers preserve the base optimizer’s Hamiltonian descent, can ensure monotonic decrease of the loss and further accelerates it. Empirically, they evaluate the proposed method first on a 2D toy objective function, then on language and vision tasks. Consistent improvements on convergence rates and training performances are shown.

**Strengths:**

1. The paper proposes cautious optimizers, a single yet effective trick for improving momentum-based gradient methods. The method is lightweight and easy to implement.
2. The authors argue the soundness of their proposed approach from a Hamiltonian/Lyapunov perspective, which might be of individual interest for future optimizer design.
3. Performance gains are shown empirically across different tasks, demonstrating the effectiveness and wide applicability of the method.

**Weaknesses:**

1. The LLM pretraining experiments use the same learning rate when comparing with original AdamW and Lion. This might not be a fair comparison since cautious optimizers use a rescaling factor $\\alpha$. Although the rescaling factor can be normalized, it is still hard to say if the same learning rate means the same thing for different optimizers.
2. The reported performance gains in Table 3 are modest, making interpretation sensitive to data and random noise. Reporting the number of random seeds tried would strengthen the claims.

Minor comments: Typo in line 390: “catuious”.

**Questions:**

Why is weight decay not included in cautious optimizers but performed after it? What if one uses other regularizers instead of the L2 norm, such as the KL regularizer in RL?

---

> ### Author Response · Authors · 2025-11-16
> **Clarifications on Hyperparameter Tuning, LM Evaluation Variance, and Reported Results**
>
> We want to first thank reviewer TurQ for their constructive feedback.
> ## Re: Same learning rates
> Table 1 is meant to highlight that Cautious’s improvement is robust and consistent across different learning rates against the baselines even without specifically tuning for Cautious (given the same hyperparameters as the baseline).
>
>
> Table 2 on the other hand removes the same hyperparameter constraints and performs discrete coordinate descent (hyperparameter search) on all hyperparameters (learning rate, weight decay, warmup steps, beta1, beta2, esp, max grad norm as well as batch size) across 4 different scales. You can find the optimal hyperparameters and search range from Appendix C (table 5 and table 6).
>
>
> We would have loved to do the same thing for Lion, but we have limited resources and decided to focus on AdamW.
>
>
> ## Re: Table 3 modest improvement/noise
> Table 3 is standard LLM downstream task evaluation done with LM Eval Harness. Taking ARC challenge as an example:
> It does not reflect randomness in model inference. ARC-Challenge is evaluated deterministically: each question is formatted as a zero-shot multiple-choice prompt, the log-likelihood of each option is computed, and the answer with the highest likelihood is selected. No sampling or stochastic decoding is used.
>
> The “±” value instead comes from the harness’s default bootstrap estimation of statistical uncertainty. Because ARC-Challenge contains only 259 questions, the harness resamples the dataset with replacement to estimate the standard deviation of the accuracy under these bootstrap draws. This quantifies finite-sample uncertainty, not inference randomness.
> In short, the evaluation procedure is fully deterministic; the variance reflects the dataset’s limited size rather than any nondeterminism in the model or evaluation pipeline.
>
> ## Re: Typos
> Thanks for pointing it out. We will make sure to fix it in later version.
>
> ## Re: Weight decay
> That is a very interesting idea. We have seen very promising results on applying similar cautious masks on top of weight decay and obtaining record results on nanogpt speedrun interacting with Muon. We did not include those results because of time constraints and the scope of the paper.
>
> Other regularization such as KL and L2 norm (equivalent to weight decay), they are orthogonal and cautious can be applied agnostically.  We have some preliminary experiments in Appendix E4, post-training small LLMs with PPO and GRPO. The initial results seem promising, though we have not had enough resources to validate further.

---

> ### Comment · Reviewer_TurQ · 2025-11-28
>
> Thank the authors for their clarifications on experiment settings and results. My evaluation remains the same.

---

### Public Comment · ~Jinghui_Yuan1 · 2026-03-03
**Spherical Cautious Optimizer – A Possible Refinement for Scale-Invariant Parameters**

Congratulations! I’m very happy to see this paper accepted. I truly appreciate the insight of this work — it is remarkably concise and effective. I am confident that this is one of the best papers I have read this year. My understanding is that the core idea is to ensure that, for each coordinate, the sign of the actual update direction $u$ should be consistent with the sign of the gradient.

I would like to further discuss the application of the cautious optimizer to scale-invariant parameters. Well-known components such as BatchNorm and LayerNorm are scale-invariant parameters [1]. Their key property is that the loss function is independent of the norm of $ w $, i.e., $ L(kw) = L(w) $. Meanwhile, the gradient lies in the tangent space of $ w $, i.e., $ g = g^\bot$. However, the update is not necessarily tangent: $u = u^\bot + u^\parallel $, where $ u^\bot $ directly decreases the loss, while $u^\parallel $ only increases the norm of $ w $. If we directly apply the cautious mask, the masking decision may be affected by the radial noise component $ u^\parallel $.

More specifically, suppose $ g_i = 1 $, $ u_i^\bot = -1 $, and the radial noise $ u_i^{\parallel} = 5 $. According to the cautious mask, this coordinate should be activated. However, the radial component $ u_i^{\parallel} $ is typically determined by preceding factors and does not provide meaningful guidance for feature learning. Therefore, for geometric consistency, we should compare $g_i = 1 $ with $ u_i^\bot = -1 $, and this coordinate should be masked out.


Therefore, for scale-invariant parameters, we attempt to project both $ u $ and $g $ onto the tangent space, and then perform element-wise comparison between $ u^\bot $ and $g^\bot$. In small-scale experiments, we observed improvements [2]. Similar to the standard cautious optimizer, this modification incurs almost no additional computational cost.

However, due to limitations in time and computational resources, we have not yet verified whether improvements persist at a  larger scale. We are currently conducting more extensive experiments to systematically evaluate this behavior. Intuitively, applying the mask along the feature learning direction — that is, the tangent direction — seems more appropriate for scale-invariant parameters.

Conceptually, this can be interpreted as a geometric continuation of your conclusion point: “(2) masking in the eigenspace rather than the parameter space.”

Given that such parameters are widely present in neural networks, I believe this discussion could be valuable. I look forward to your thoughts.

[1] Spherical motion dynamics: Learning dynamics of neural network with normalization, weight decay, and sgd

[2] Spherical Cautious Optimizer [https://openreview.net/forum?id=OyT2CJ4fh7]

---
In addition, there appears to be a minor typo in Algorithm 2: Cautious AdamW (C-AdamW)

On line 13 “// Add weight decay”, the coefficient of $ w_t $ should be $ \epsilon \gamma $.

---

### Meta-Review · Area_Chair_aFq7 · 2026-01-08

**Summary:**

This work proposes to modify updates from existing optimizers by masking directions not aligned with the gradient. They show a convincing theoretical justification and empirical results with LLM model training. The reviewers were generally positive about the work. Some concerns regarding the empirical evaluation and the limits of the theory, which was addressed in the rebuttal. Overall  the benefits outweigh the weakness.

**Reviewer Concerns:**

The following were concerns that were addressed in the rebuttal
-- Fairness of evaluation
-- Gap between theory and implementation
-- Scale of the experiments

**Reviewer Scores:**

4ez6 indicated they would increase their scores, other reviewers seemed leaning towards maintaining their scores

---

### Decision · Program_Chairs · 2026-01-26

Accept (Poster)